# Fast three-color single-molecule FRET using statistical inference

Janghyun Yoo[1,2], Jae-Yeol Kim [1,2], John M. Louis[1], Irina V. Gopich[1] & Hoi Sung Chung [1✉]

We describe theory, experiments, and analyses of three-color Förster resonance energy transfer (FRET) spectroscopy for probing sub-millisecond conformational dynamics of protein folding and binding of disordered proteins. We devise a scheme that uses single continuous-wave laser excitation of the donor instead of alternating excitation of the donor and one of the acceptors. This scheme alleviates photophysical problems of acceptors such as rapid photobleaching, which is crucial for high time resolution experiments with elevated illumination intensity. Our method exploits the molecular species with one of the acceptors absent or photobleached, from which two-color FRET data is collected in the same experiment. We show that three FRET efficiencies and kinetic parameters can be determined without alternating excitation from a global maximum likelihood analysis of two-color and three-color photon trajectories. We implement co-parallelization of CPU-GPU processing, which leads to a significant reduction of the likelihood calculation time for efficient parameter determination.

[1] Laboratory of Chemical Physics, National Institute of Diabetes and Digestive and Kidney Diseases, National Institutes of Health, Bethesda, MD 20892-0520, USA. [2] These authors contributed equally: Janghyun Yoo, Jae-Yeol Kim. ✉email: chunghoi@niddk.nih.gov

Single-molecule Förster resonance energy transfer (FRET) spectroscopy is a very sensitive tool to detect distance changes on a nanometer scale. Since its development, it has become an indispensable experimental method in many areas of modern biological sciences[1–6]. In most single-molecule FRET studies, two-color FRET with a single pair of fluorophores (i.e., one donor and one acceptor) is used to deduce molecular states from the single distance information. Three-color FRET can provide much more information on molecular conformations[7,8]. For example, by attaching three fluorophores to a protein or nucleic acid, three-dimensional (3D) conformational changes including correlated motions of different parts of a molecule can be followed during folding[7,9]. Three-color FRET can also be used for probing molecular interactions. If conformational changes occur during protein-protein interactions, such as coupled binding and folding of intrinsically disordered proteins (IDPs)[10], by attaching two fluorophores to a disordered protein and the third fluorophore to the second molecule, it is possible to correlate conformational changes of the IDP and its interaction with a binding partner. These processes often occur on a fast time scale (µs–ms)[9,11–14], but the time resolution of typical three-color FRET experiments remains at tens of milliseconds. In this article, we describe our development of three-color FRET spectroscopy using intense continuous-wave (CW) laser excitation for the investigation of fast molecular processes occurring on µs to ms time scales.

Figure 1 provides a summary of the method in this paper. Typically three-color FRET is done with alternating laser excitation (ALEX)[15,16] using CW lasers with intensity modulations or pulse-interleaved excitation (PIE)[17] using pulsed lasers to determine all three FRET efficiencies (Fig. 1b)[18,19]. However, additional acceptor excitation increases photophysical problems such as photoblinking and photobleaching particularly when using intense pulses to collect data at a high photon detection rate (e.g., >50 ms$^{-1}$). Single CW excitation alleviates these photophysical problems (see Supplementary Fig. 1 for the comparison of photophysical properties in single CW excitation and PIE). To determine all three FRET efficiencies by single CW laser excitation, we globally analyze three-color and two-color parts of trajectories (Fig. 1c, d). Due to the incomplete labeling and photobleaching of one of the two acceptors during the measurement, two-color segments are always present and can be utilized. By determining one FRET efficiency from the two-color photon trajectories, it is possible to calculate all three FRET efficiencies as previously demonstrated for slower processes[20,21]. To enhance the time resolution, we extended the photon trajectory analysis using the maximum likelihood method developed by Gopich and Szabo[22] to the three-color analysis. This global

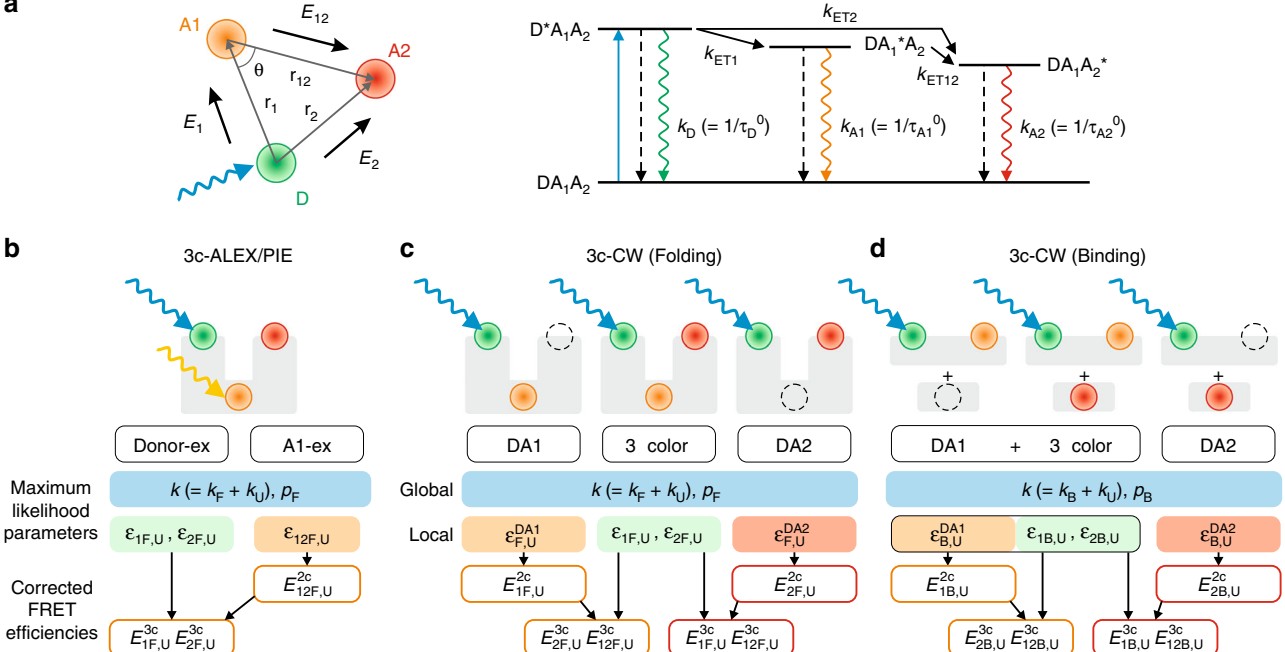

**Fig. 1 Three-color FRET. a** When the donor (D) is excited by a laser (blue arrows), the excited energy decays to the ground state either through the radiative (rippled arrows) or non-radiative pathways (dashed arrows) with rate constant $k_D$ or the excited energy is transferred to A1 and A2 with transfer rates $k_{ET1}$ and $k_{ET2}$, respectively. The excited state energy of A1 can also be transferred to A2 with a transfer rate $k_{ET12}$. The two-color FRET efficiencies ($E_1$, $E_2$, and $E_{12}$) are the efficiencies of the energy transfer between corresponding two dyes in the absence of a third dye. **b** Three-color alternating laser excitation or pulse-interleaved excitation (3c-ALEX/PIE) scheme. A molecule (light gray) containing all three fluorophores is excited by a donor excitation laser (Donor-ex) and an A1 excitation laser (yellow rippled arrow, A1-ex) alternately. The parameters are determined from the global analysis of the photon trajectories. The relaxation rate $k$ and the population of the folded state $p_F$ are global fitting parameters. The fractions of acceptor 1 ($\varepsilon_1$) and acceptor 2 ($\varepsilon_2$) photon count rates in the folded (F) and unfolded (U) states are determined from the photon trajectories by D excitation. From the trajectories by A1 excitation, the fraction of A2 count rate, $\varepsilon_{12}$, is determined. These acceptor fractions are converted to FRET efficiencies defined in (**a**) (see Methods). **c** The global analysis scheme of three- and two-color segments collected in a three-color CW donor excitation (3c-CW) folding experiment. From the two-color acceptor fractions, $\varepsilon^{DA1}$ and $\varepsilon^{DA2}$, two-color FRET efficiencies $E_1^{2c}$ and $E_2^{2c}$ are obtained, respectively. Using these values, two sets of three-color FRET efficiencies can be calculated and compared. **d** The global analysis scheme in a three-color CW donor excitation (3c-CW) binding experiment. The three-color and DA1 parts of the photon trajectories cannot be separately analyzed as in (**c**) because the unbound state is always two-color (DA1), whereas the bound state is either three-color or two-color (DA1) due to the binding to either A2-labeled (middle) or A2-unlabeled (left) binding partner. The three-color FRET efficiencies can be determined similarly using the acceptor fractions obtained from the maximum likelihood method.

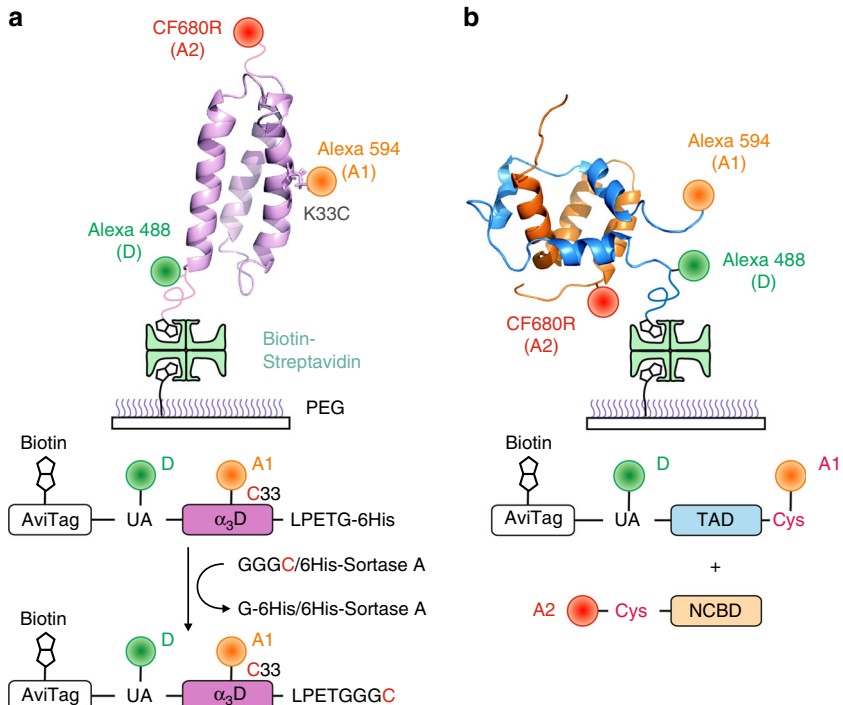

**Fig. 2 Three-color FRET in protein folding and binding. a** Three different fluorophores are attached to $\alpha_3 D$ site-specifically. Alexa 488 (donor, D) and Alexa 594 (acceptor 1, A1) are attached to 4-acetylphenylalanine (UA) at the N-terminus and cysteine at residue 33, respectively. Then, a cysteine residue is attached to the C-terminus using sortase-mediated ligation of a short peptide GGGC. This cysteine residue is labeled with CF680R (acceptor 2, A2). The labeled protein is immobilized on a polyethyleneglycol (PEG)-coated glass surface using biotin-streptavidin linkage. In two-color FRET, site-specific labeling of two fluorophores is not usually necessary because the two species with the different donor and acceptor positions would not cause a significant difference in the FRET efficiency unless fluorescence quenching occurs at one of the two labeling positions. In three-color FRET, however, when A1 and A2 are switched, for example, the distances between D and A1 and between D and A2 become different. **b** D and A1 are attached to 4-acetylphenylalanine and cysteine residues at the N- and C-termini of the transactivation domain (TAD), respectively, which is immobilized on the surface. TAD is incubated with A2-labeled NCBD in solution at a concentration close to the dissociation constant to monitor binding and dissociation events.

analysis of fluorescence photon trajectories using the maximum likelihood method determines many parameters simultaneously and therefore involves extensive computation. We developed a fitting routine using CPU-GPU co-parallelization that reduces the computation time significantly, by almost two orders of magnitude compared to the calculation with a single CPU processor (see Supplementary Fig. 2 and "Methods"). (Analysis codes are available at https://github.com/hoisunglab/FRET_3colorCW.) We also describe the correction procedures to obtain accurate FRET efficiencies that can be related to the distances between the dyes.

To demonstrate the performance of the three-color CW method, we apply this method to two different systems: (1) folding of a designed protein, $\alpha_3 D$[23–25], and (2) binding of a disordered protein, the N-terminal transactivation domain (TAD) of the tumor suppressor protein p53, and one of its binding partners, the nuclear coactivator binding domain (NCBD) of the CREB binding protein (CBP)[26] (Fig. 2). All fluorophores are attached site-specifically[13,18] (see Methods and Fig. 2). Although both systems are two-state systems, the data of the binding experiment is much more complex and a completely different kinetics model is used in the analysis. In both experiments, we successfully extract fast ($\mu$s–ms) kinetic parameters, which are similar to the previous two-color experimental results. The analysis of simulated photon trajectories shows that folding and binding rates close to the photon count rate (65 ms$^{-1}$) can be successfully extracted. We also compare the FRET efficiencies obtained from three-color measurements and two-color measurements and discuss the discrepancies between these values. Importantly, we find that the FRET efficiency conversion from

the extracted fractions of acceptor photon count rates (acceptor fraction, $\varepsilon$) can be inaccurate depending on the specific set of values of a system. In this case, it is important to choose a FRET efficiency conversion scheme that leads to the smallest errors to calculate the three-color FRET efficiencies.

## Results

**Design of protein constructs and dye labeling**. Site-specific labeling is necessary in three-color experiments to avoid ambiguity resulting from molecular species with different combinations of labeling positions (see Fig. 2a). In the experiment of $\alpha_3 D$ folding, we attached three fluorophores site-specifically and the protein was immobilized on a surface as shown in Fig. 2a[18]. Folding and unfolding were monitored at three different guanidinium chloride (GdmCl) concentrations (2, 2.25, and 2.5 M) near the denaturation mid-point. In the binding experiment, disordered TAD was labeled with two fluorophores (donor and A1) site-specifically, immobilized, and incubated with NCBD labeled with A2. TAD folds upon binding NCBD in solution. Binding and dissociation were monitored at the concentration of NCBD near the dissociation constant at three different NaCl concentrations (0, 10, and 30 mM).

**Fraction of acceptor photon count rate and FRET efficiency**. The parameters extracted using the maximum likelihood analysis are not the FRET efficiencies, but the fractions of acceptor photon count rates. Specifically, $\varepsilon_1$ and $\varepsilon_2$ are the fractions of A1 and A2 in the three-color photon trajectories (see Eq. (2)), and $\varepsilon^{DA1}$ and

$\varepsilon^{DA2}$ are the fractions of A1 and A2 in the two-color DA1 and DA2 trajectories, respectively (see Eq. (3)).

The acceptor fractions were determined by global analysis of the two- and three-color parts of trajectories using the maximum likelihood method with appropriate kinetic models (Fig. 1c, d). We have shown previously that the kinetic parameters, especially the rate coefficients on the time scale of milliseconds and shorter are affected by acceptor blinking[27]. To obtain more accurate parameters, blinking of both acceptors was incorporated into the models. As a result, we used eight-state models both for folding of $\alpha_3D$ and for TAD/NCBD binding (see Supplementary Fig. 3 and Methods for the details of the kinetic models used in this study). The extracted acceptor fraction parameters were converted to the FRET efficiencies (see Methods).

The three FRET efficiencies between the three fluorophores are defined in Fig. 1a and Methods. $E_1$, $E_2$, and $E_{12}$ are the FRET efficiencies between D and A1, between D and A2, and between A1 and A2 in the absence of A2, A1, and D, respectively. These values are directly related to the distances between two corresponding fluorophores. We refer to the FRET efficiencies obtained directly from the two-color parts of trajectories, DA1 and DA2, as $E_1^{2c}$ and $E_2^{2c}$, respectively, and the FRET efficiencies from the three-color part as $E_1^{3c}$, $E_2^{3c}$, and $E_{12}^{3c}$. These three-color FRET efficiencies can be found in two ways, using $E_1^{2c}$ (Eq. (7)) or $E_2^{2c}$ (Eq. (8)).

**CPU–GPU co-parallelization for the likelihood calculation**. The calculation of the likelihood function requires a large number of matrix–vector multiplications. To accelerate the optimization, we parallelized the likelihood calculation using both GPU and CPU (Supplementary Fig. 2a). In the analysis, a photon trajectory of one molecule is separated into different color segments (three-color, DA1, and DA2 for $\alpha_3D$ folding and three-color/DA1 and DA2 for TAD/NCBD binding). The basic idea is to parallelize the evaluation of the likelihood values of independent segments of photon trajectories by distributing these calculations to all available GPU and CPU processors (see Methods). The acceleration by parallel processing are compared in Supplementary Fig. 2.

**Chemical denaturation of $\alpha_3D$**. Representative equilibrium fluorescence trajectories (1 ms bin time) are shown in Fig. 3a for folding and unfolding of $\alpha_3D$. The upper trajectory begins with a three-color segment where all three dyes are active (i.e., fluorescing), which is followed by a two-color segment where D and A1 are active (DA1 segment) after A2 photobleaching (red arrow) and a donor-only segment after subsequent A1 photobleaching (orange arrow). In the lower trajectory, A1 is photobleached first, which leads to the transition from a three-color segment to a segment containing active D and A2 (DA2 segment). Although previous two-color single-molecule experiments show that $\alpha_3D$ is a two-state protein[25], folding and unfolding transitions are not readily observable in the binned trajectory in Fig. 3a. Histograms of $\varepsilon_1$ and $\varepsilon_2$ for the three-color segments (Fig. 3b) and $\varepsilon$'s for DA1 and DA2 (Fig. 3c) at three different GdmCl concentrations also show almost featureless single peaks. However, $\varepsilon_2$ histogram at 2.25 M shows two peaks with a significant overlap, suggesting there are two states. In addition, $\varepsilon_2$ peaks at 2 and 2.5 M are skewed toward the higher and lower side of the two peaks at 2.25 M GdmCl, respectively, indicating the shift of the equilibrium from the folded (high $\varepsilon_2$) to the unfolded (low $\varepsilon_2$) state by denaturation. When the protein unfolds, all three distances between the fluorophores increase. Therefore, the decrease of $\varepsilon_2$ with the increasing GdmCl concentration is an expected result. On the other hand, the change in $\varepsilon_1$ is more complex. When the

protein unfolds, $\varepsilon_1$ can decrease because of less transfer from D to A1 ($E_1$), whereas it can also increase because of less transfer from A1 to A2 ($E_{12}$). The increase in $\varepsilon_1$ with the increasing GdmCl concentration suggests that the net effect is the detection of more A1 photons in the unfolded state. The shifts of the histograms of two-color trajectories (DA1 and DA2, Fig. 3c) are much smaller primarily due to the small difference of acceptor fractions $\varepsilon^{DA1}$ and $\varepsilon^{DA2}$ between the folded and unfolded states (see accurate determinations of parameters using the maximum likelihood method in the following sections).

We also performed free-diffusion experiments, in which molecules are not immobilized, but freely diffuse and emit a burst of fluorescence photons when they pass through the laser focus. The one- and two-dimensional acceptor fraction histograms from the free-diffusion experiments are similar to those from the immobilization experiments (Supplementary Fig. 4), which indicates the surface immobilization effect on the protein dynamics is small as also verified in the previous two-color FRET study[25].

**Coupled binding and folding of TAD**. As mentioned in Introduction, binding of TAD and NCBD is a two-state system as $\alpha_3D$ folding, but the data is much more complex as shown in the trajectories in Fig. 3d and a different kinetic model needs to be used. In the upper trajectory in Fig. 3d, which was collected at 0 mM NaCl, the binding and dissociation kinetics are relatively slow and states can be distinguished by visual identification of binding and dissociation transitions as indicated by different colors in the bar above the trajectory. The trajectory begins with the unbound state, where A2 intensity is close to the background level and the donor intensity is higher than A1 intensity. In the second fragment (magenta color in the bar), A2 intensity is the highest and both donor and A1 intensities are low, indicating this is the bound state with three active fluorophores. Therefore, the transition from the first to the second fragment results from binding of an A2-labeled NCBD molecule. The next short fragment is the unbound state again, and the transition to this fragment is dissociation. This binding and dissociation transitions repeat in the following fragments. Then, in the middle of the trajectory, different transitions are observed as indicated by cyan–orange–cyan in the color bar. The orange fragment is a two-color fragment as the unbound state, but A1 intensity is higher than D intensity, indicating the FRET efficiency $E_1$ is higher than that of the unbound state. Therefore, this fragment corresponds to the bound state with an unlabeled (or with inactive A2) NCBD molecule. Since the photons emitted by three (DA1A2, bound state) and two (DA1, both bound and unbound state) active fluorophores are mixed (Fig. 3d, Supplementary Fig. 3), they must be analyzed together (i.e., one kinetic model) without breaking into independent segments (3-color/DA1 segment).

In addition to binding and dissociation transitions, photophysical transitions are also observed in the trajectory. For example, at ~200 ms in the upper trajectory, a fragment of the two-color bound state (orange) is immediately followed by a three-color bound state fragment (magenta). It is possible that a very short unbound state is present between these two bound states, but it is more likely that this is a transition from the A2 dark state to A2 bright state. Since A2-labeled NCBD molecules diffuse in the focal volume for a while before binding, direct A2 excitation can cause the A2 dark state before binding. The reverse transitions are also observed in other trajectories, which can be attributed to blinking or photobleaching of A2. Since these relatively slow photophysical transitions cannot be excluded from the analysis, these should also be incorporated in the kinetic

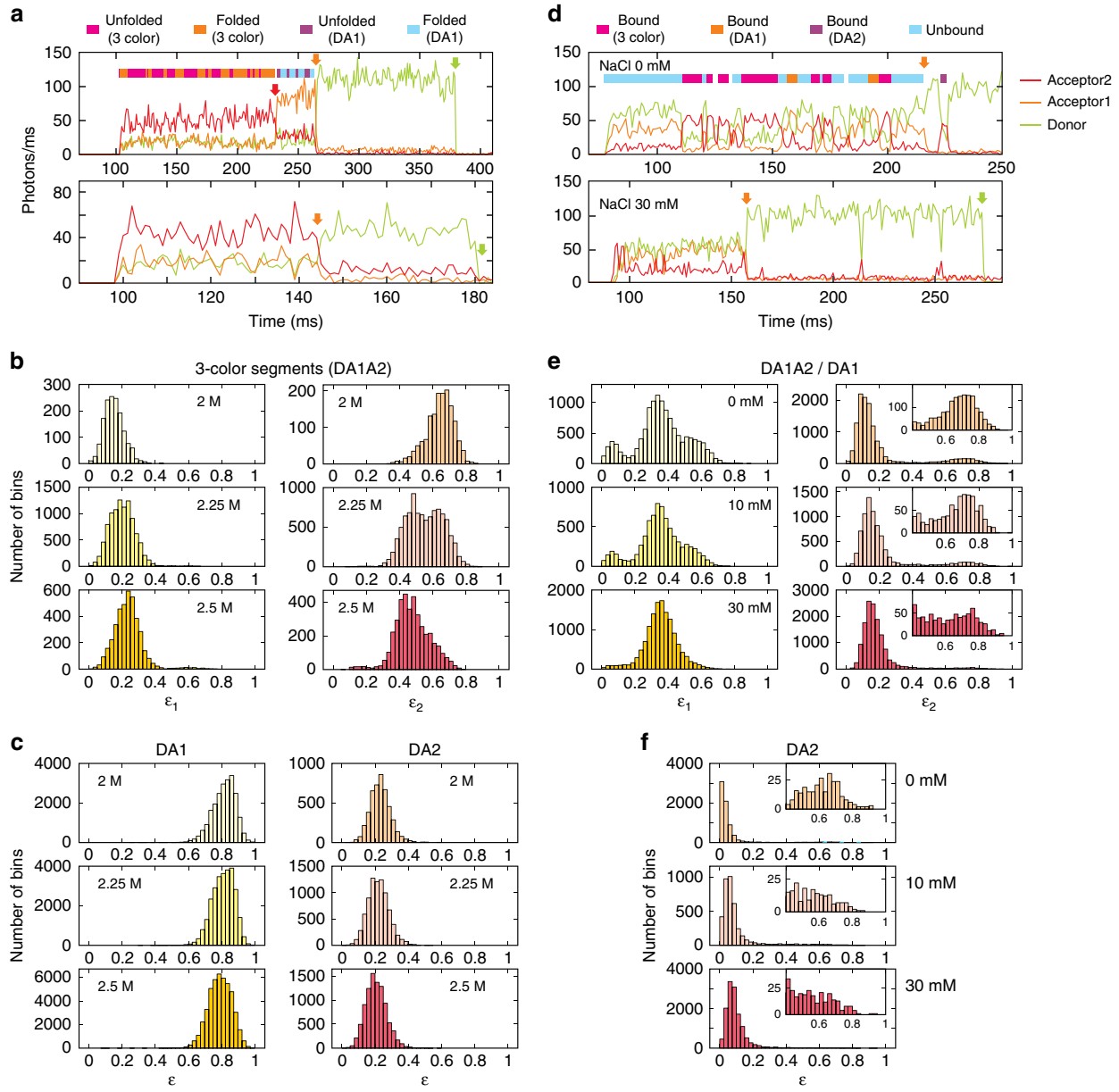

**Fig. 3 Fluorescence trajectories and histograms of fraction of acceptor photon counts.** The bin time of 1 ms is used for the trajectories and histograms. Green, orange, and red arrows indicate photobleaching of D, A1, and A2, respectively. **a–c** Fluorescence trajectories and acceptor photon fraction histograms of $\alpha_3D$ at different GdmCl concentrations. 107 (2 M), 195 (2.25 M), and 215 (2.5 M) three-color DA1A2 trajectories (i.e., molecules), 40 (2 M), 190 (2.25 M), and 212 (2.5 M) two-color DA1 trajectories, and 84 (2 M), 110 (2.25 M), and 188 (2.5 M) two-color DA2 trajectories were analyzed. **a** Two $\alpha_3D$ trajectories of the donor (green), acceptor 1 (orange) and acceptor 2 (red) fluorescence at 2.25 M GdmCl. In the upper trajectory, A2 fluorescence after A2 photobleaching (red arrow) that is maintained before A1 photobleaching (orange arrow) results from the leak of A1 photons into A2 channel. States assigned by the Viterbi algorithm are shown in the color bar above the upper trajectory. **b** Histograms of the uncorrected fractions of A1 ($\varepsilon_1$) and A2 ($\varepsilon_2$) from three-color segments (DA1A2). **c** Histograms of the uncorrected fractions of acceptor photons ($\varepsilon$) from two-color segments (DA1 and DA2). **d–f** Fluorescence trajectories and acceptor photon fraction histograms of TAD-NCBD binding at different NaCl concentrations. 140, 140, and 186 molecules were analyzed at 0, 10, and 30 mM NaCl, respectively. **d** Donor (green), acceptor 1 (orange) and acceptor 2 (red) fluorescence trajectories at 0 mM (upper) and 30 mM NaCl (lower). Apparent states are assigned by visual inspection as indicated in the color bar above the trajectory at 0 mM. States assigned by the Viterbi algorithm for a trajectory at 30 mM NaCl are shown in Supplementary Fig. 5. **e** Histograms of the uncorrected fractions of A1 ($\varepsilon_1$) and A2 ($\varepsilon_2$) from three-color (DA1A2)/DA1 segments. Enlarged histograms of the bound state ($0.4 < \varepsilon_2 < 1$) are shown in the insets. **f** Histograms of the uncorrected fractions of acceptor photons from DA2 segments. Enlarged histograms of the bound state ($0.4 < \varepsilon < 1$) are shown in the insets.

model for the maximum likelihood analysis (see the next section, Supplementary Fig. 3, and Methods for kinetic models).

Unlike three-color/DA1 segments, DA2 segments are separable. For example, DA2 segment appears after acceptor 1 photobleaching (orange arrow in Fig. 3d). In DA2 seg-ments, the A2 dark state is indistinguishable from the unbound state. Therefore, the A2 dark state should be included in the kinetic model with transition rates and dark state population as global parameters (see Supplementary Fig. 3d).

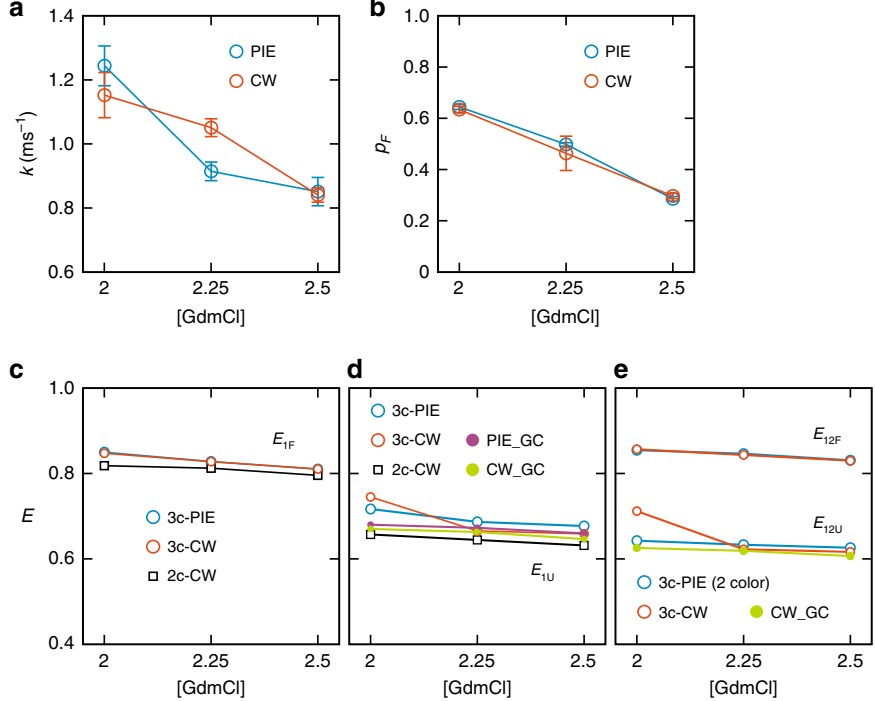

**Fig. 4 Extracted parameters of α₃D folding from 3c-PIE and 3c-CW. a** Relaxation rate $k$ ($= k_F + k_U$) from 3c-PIE (blue) and 3c-CW (red). **b** Fraction of the folded state $p_F$ from 3c-PIE (blue) and 3c-CW (red). **c** Folded state FRET efficiency $E_{1F}$ from two color (black), 3c-PIE (blue), and 3c-CW (red). **d** Unfolded state FRET efficiency $E_{1U}$ from two color (black), 3c-PIE (blue), 3c-CW (red), and $E_{1U}^{3c}$ calculated using the Gaussian chain model with parameters obtained from two-color FRET efficiency for 3c-CW (green) and 3c-PIE (purple) (see Methods). In **c**, **d**, $E_1^{3c}$ values of 3c-CW were determined using $E_2^{2c}$ from two-color (DA2) segments. $E_1^{3c}$ values of 3c-PIE were calculated using $E_{12}^{2c}$ obtained from A1 excitation. **e** FRET efficiencies between A1 and A2 in the folded ($E_{12F}^{2c}$) and unfolded ($E_{12U}^{2c}$) states in 3c-CW using two-color $E_2^{2c}$ (red), the unfolded state FRET efficiency calculated using the Gaussian chain model (green), and $E_{12F}^{2c}$ and $E_{12U}^{2c}$ determined from 3c-PIE by A1 excitation (blue). Error bars are standard deviations calculated from the curvature at the maximum of the likelihood function. Source data are provided as a Source Data file.

As the salt concentration is increased, the binding and dissociation kinetics become faster and visual identification of transitions is not possible anymore as shown in the lower trajectory (30 mM NaCl) in Fig. 3d. This effect can be better visualized in the histogram in Fig. 3e. At 0 mM, there are three peaks in the $\varepsilon_1$ histogram of three-color/DA1 segments. The peak at $\varepsilon_1$ ~0.08 is the bound state with an active A2 (most of the excited energy of A1 is transferred to A2). The peak at $\varepsilon_1$ ~0.6 corresponds to the bound state without an active A2. The peak at $\varepsilon_1$ ~0.33 corresponds to the unbound state. On the other hand, at 30 mM NaCl, the bound state peaks disappear and are merged to the unbound state peak due to the faster binding and dissociation kinetics. In this case, the parameters can be extracted only by the maximum likelihood analysis of photon trajectories.

Similar features are observed in the $\varepsilon_2$ histograms of three-color/DA1 segments (Fig. 3e) and the histograms of DA2 segments (Fig. 3f). In these histograms, the large peak at lower $\varepsilon$ is the unbound state and the small peak at high $\varepsilon$ (see the insets) corresponds to the bound state (there are less data in the bound state due to low labeling efficiency and photobleaching of A2). At 0 mM, there is a peak at ~0.7 and ~0.65 in $\varepsilon_2$ histograms of the three-color/DA1 and DA2 segments, respectively. At 30 mM, the bound state peaks become broad and smear toward the large unbound state peak at lower $\varepsilon$ due to the faster kinetics.

The two-dimensional acceptor fraction histograms from the free-diffusion and immobilization experiments are shown in Supplementary Fig. 4. Due to the low $\gamma$-factor of A2 ($\gamma_2 = 0.49$), the bound state peak is not observed in the free-diffusion histograms and a direct comparison with the immobilization data is not possible. However, the dissociation constant measured in

this work is similar to that from an ensemble binding experiment, which indicates the surface immobilization effect is very small (see below for more discussion).

**FRET efficiencies and kinetic parameters of α₃D folding**. Figure 4 shows the relaxation rates, fraction of the folded state, and corrected FRET efficiencies of α₃D folding at three different GdmCl concentrations, which were extracted using the maximum likelihood method with the eight-state model including acceptor blinking (Supplementary Fig. 3c). The parameters are also listed in Supplementary Table 1. The relaxation rates of ~1 ms$^{-1}$ (Fig. 4a) and folded state populations (Fig. 4b) are consistent with those of the previous measurements using two-color FRET[25] and three-color PIE experiments[18]. The histograms constructed from recolored photon trajectories using maximum likelihood parameters (eight-state model)[22] are very similar to the experimental histograms (Supplementary Fig. 6), confirming the good accuracy of the extracted parameters. The cross-correlation analysis of photon trajectories also shows the relaxation rates (2.25 M GdmCl) are comparable to that from the maximum likelihood method (Supplementary Fig. 6e).

Figure 4c, d and Supplementary Fig. 7 compare the FRET efficiencies $E_1$ and $E_2$ of the folded and unfolded states obtained from (1) the three-color segments by CW excitation (3c-CW), (2) the two-color segments (2c-CW), and (3) the previous three-color PIE (3c-PIE) experiment[18]. As mentioned above, in 3c-PIE, three-color values $E_1^{3c}$ and $E_2^{3c}$ are determined using a two-color value $E_{12}^{2c}$ (Eq. (6)). In 3c-CW, there are two ways: $E_2^{3c}$ and $E_{12}^{3c}$ are determined using $E_1^{2c}$ (Eq. (7)) or $E_1^{3c}$ and $E_{12}^{3c}$ are determined

using $E_2^{2c}$ (Eq. (8)). The three-color FRET efficiencies of the folded (F) and unfolded (U) states, $E_{2F}^{3c}$ and $E_{2U}^{3c}$, in both 3c-PIE and 3c-CW (determined using $E_1^{2c}$) measurements vary largely with GdmCl concentration, whereas the two-color FRET efficiencies $E_{2F}^{2c}$ and $E_{2U}^{2c}$ in the 2c-CW measurement are stable (Supplementary Fig. 7). The inaccuracy of these values in 3c-PIE and 3c-CW results from the large propagation error in the determination of $E_2^{3c}$, which is unavoidable because this happens when $E_2$ is low (see Supplementary Fig. 8 for the propagation of the error of the two-color FRET efficiency values to the determination of the three-color FRET efficiencies). This large uncertainty suggests that a two-color value $E_2^{2c}$ should be used to determine three-color FRET efficiencies $E_1^{3c}$ and $E_{12}^{3c}$ when $E_2$ is low (Supplementary Table 1).

Compared to $E_2^{3c}$, $E_1^{3c}$ values are stable. The folded state FRET efficiency $E_{1F}$ from all three measurements agree very well (Fig. 4c). These values are also close to the FRET efficiency of 0.81 calculated from the distance distribution of D and A1 attached to residue 1 and 33 obtained from the accessible volume calculation[28] using the protein structure (PDB id: 2A3D, Fig. 2). On the other hand, there are discrepancies between the unfolded state FRET efficiency values, $E_{1U}^{2c}$ and $E_{1U}^{3c}$ of both 3c-PIE and 3c-CW (Fig. 4d). A part of this discrepancy results from the fluctuating distance between the dyes because the unfolded protein is flexible. The effect of fluctuating distances has been explained and rigorously evaluated previously using the Gaussian chain model[18] (see Methods). However, the effect of fluctuating distance evaluated by the Gaussian chain model for the FRET efficiency values of $\alpha_3D$ is relatively small, although it can be large in general (see Supplementary Fig. 9). This indicates that the difference between the experimental $E_{1U}^{3c}$ and $E_{1U}^{2c}$ values may result from the measurement error. In ref. [18], we have shown that $E_{1U}^{3c}$ values calculated using the Gaussian chain model agree very well with the 3c-PIE values, but this resulted from the usage of $\langle E_{1+2} \rangle$ value calculated from three-color $E_{1U}^{3c}$ and $E_{2U}^{3c}$ rather than two-color $E_{1U}^{2c}$ and $E_{2U}^{2c}$ in Eq. (11) in ref. [18]. The correct values are plotted in Fig. 4d, which shows small differences between two- and three-color $E_{1U}$ values.

The comparison of $E_{12}$ shows similar trends (Fig. 4e). For the folded state, $E_{12F}^{3c}$ in 3c-CW calculated using $E_{2F}^{2c}$ coincide with the values from 3c-PIE experiment (i.e. two-color values $E_{12F}^{2c}$ by A1 excitation). For the unfolded state FRET efficiency, $E_{12U}^{3c}$, although there is a deviation at 2 M GdmCl, the values of the other two concentrations agree well with $E_{12U}^{2c}$ from 3c-PIE experiment. These values are also very similar to the three-color $E_{12U}^{3c}$ calculated using the Gaussian chain model.

**FRET efficiencies and kinetic parameters of TAD/NCBD binding**. Figure 5 shows the association and dissociation rate coefficients, dissociation constant ($K_d$), and FRET efficiencies of TAD/NCBD binding at three different NaCl concentrations, which were obtained using the maximum likelihood method with the eight-state model including acceptor blinking (Supplementary Fig. 3f). The parameters are also listed in Supplementary Tables 2 and 3. Similar to $\alpha_3D$ experiment, the histograms constructed from the recolored photon trajectories using the extracted parameters (eight-state model) agree well with the experimental histograms (Supplementary Fig. 6). Compared to the previous two-color FRET results of binding of Alexa 488- and Alexa 647-labeled TAD and unlabeled NCBD[13], the association rate coefficients are smaller by a factor of 2, but still in the regime of diffusion-limited association. The dissociation rates from the three-color measurement at 0 and 10 mM are 2–3 times faster

than those from the two-color measurement, whereas the dissociation rates at 30 mM are comparable in both experiments, indicating the bound complex is destabilized at low ionic strength. The different stability of the bound complex may result from different labels and labeling positions in three-color and two-color experiments. Nevertheless, the dissociation constant $K_d$ from our single-molecule measurements are comparable with those from ensemble measurements (isothermal calorimetry) by Jemth and coworkers[29]. At the ionic strength of 44 and 74 mM, the dissociation constants were 3 and 13 μM, which agree very well with our previous two-color measurement ($K_d = 1.6$–11 μM) at similar ionic strength of 38–98 mM[13]. In addition, the strong ionic strength dependence of both association and dissociation is still observed, suggesting electrostatic interactions play a similarly important role[13].

Figure 5c, d compare the three-color and two-color FRET efficiencies, $E_1$ and $E_2$. In the unbound state, only donor and acceptor 1 emit photons. Therefore, only two-color FRET efficiencies are presented for $E_{1U}$ and $E_{2U}$. The FRET efficiencies in the unbound state, $E_{2U}$ and $E_{12U}$, should be 0 after corrections for background and D and A1 leaks, which is the case as shown in Fig. 5c, d. $E_{1U}$ increases slightly with the increasing NaCl concentration because of the collapse of the disordered TAD due to the increased screening effect of electrostatic repulsion (net charge is −10) at high ionic strength, consistent with the previous two-color FRET result[13]. For the bound state, both FRET efficiencies, $E_{1B}$ and $E_{2B}$, obtained from the two-color and three-color segments are very similar at 0 mM NaCl. However, they differ by ~0.1 at 10 and 30 mM NaCl. This discrepancy may imply the presence of conformational flexibility in the bound state. Although several residues of TAD at both N- and C-termini are disordered in the NMR structure of the bound complex[26], the average distance between the dyes is not expected to be affected significantly by the ionic strength. Instead, the discrepancy between the two-color and three-color FRET efficiencies of the TAD/NCBD bound complex results more likely from the measurement error because there is no consistent trend in the differences of the extracted parameters. For example, $E_{1B}^{2c}$ is the lowest at 30 mM (0.71), whereas $E_{2B}^{2c}$ is the lowest at 10 mM. $\varepsilon_2$, is the lowest at 0 mM (Supplementary Table 2). These results suggest that a combination of the variations in the values listed above can cause the deviation of the two- and three-color FRET efficiencies. Nonetheless, the difference of $E_{12B}$ values obtained from the two conversion methods using $E_{1B}^{2c}$ or $E_{2B}^{2c}$ is relatively small (Fig. 5d).

**Determination of microsecond kinetic parameters**. The kinetics of the experimental systems that we studied are near 1 ms$^{-1}$ or slightly faster. To verify the applicability of the method to the analysis of faster kinetics, we performed the analysis of simulated photon trajectories. The experimental photon trajectories at 2–2.5 M GdmCl for $\alpha_3D$ folding and 0 mM NaCl for TAD/NCBD binding were recolored[22] using experimental maximum likelihood parameters with varying relaxation rate and folded (or bound) population. The recolored photon trajectories were analyzed again the same way as the experimental trajectories. Figure 6 shows that the determined relaxation rates and relative populations are very accurate even when the relaxation rate is close to the average photon count rate (~80 ms$^{-1}$ for $\alpha_3D$ folding and 65 ms$^{-1}$ for TAD/NCBD binding) (see Supplementary Fig. 10 for the extracted acceptor fractions). This simulation result indicates that the dynamic range of the three-color maximum likelihood method is much wider than the experimental results shown in this work as examples.

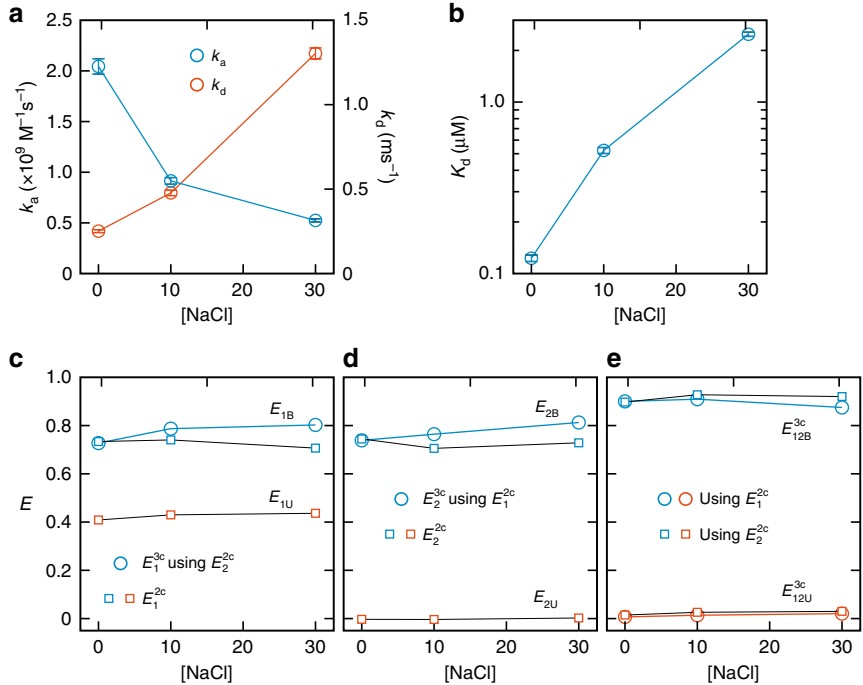

**Fig. 5 FRET efficiencies and kinetics parameters of TAD–NCBD binding. a** Bimolecular association rate constant $k_a = k_B/[\text{NCBD}]$ (blue) and dissociation rate constant $k_U$ (red) obtained from the extracted relaxation rate, $k = k_B + k_U$, and the bound fraction, $p_B = k_B/k$. $k_B$ is the apparent binding rate constant. **b** Dissociation constant, $K_d$ $(= k_U/k_a)$. **c–e** Measured FRET efficiencies of the bound and unbound states. **c** $E_1^{2c}$ (blue square, bound; red square, unbound) and $E_{1B}^{3c}$ (blue circle) calculated using $E_{2B}^{2c}$. **d** $E_2^{2c}$ (blue square, bound; red square, unbound) and $E_{2B}^{3c}$ (blue circle) calculated using $E_{1B}^{2c}$. **e** $E_{12}^{3c}$ (blue, bound; red, unbound) calculated using two-color FRET efficiencies $E_1^{2c}$ (circle) and $E_2^{2c}$ (square). Error bars are standard deviations calculated from the curvature at the maximum of the likelihood function. Source data are provided as a Source Data file.

**Detection of intermediate state**. We also tested the applicability of the method to the detection of an intermediate state with relatively low population (10%) and short lifetime. Photon trajectories of TAD/NCBD binding at 0 mM NaCl were simulated by recoloring experimental trajectories using the three-state model for binding at various bound state population and rates (Supplementary Fig. 11). The simulated trajectories were analyzed with two- and three-state models for comparison. The fraction of the intermediate state and kinetic parameters can be extracted with reasonable accuracy as shown in Supplementary Fig. 11.

When the number of states is unknown, it is usually determined by comparing statistical criteria such as Bayesian information criterion (BIC). In Supplementary Fig. 11h, BIC is clearly smaller for the three-state model compared to the two-state model. However, one should be cautious to use statistical criteria such as BIC to determine the number of states. In real experimental data, there are various factors that can cause the system look more complex (i.e., more states) such as heterogeneity in the immobilization microenvironment and fluorescence signal from impurity. Therefore, it would be prudent to use any prior knowledge from other experimental methods rather than blindly trust statistical criteria from the analysis.

## Discussion

In this work, we have described our development of fast three-color single-molecule FRET experiment and photon trajectory analysis methods to determine FRET efficiencies and μs–ms timescale kinetics of folding of $\alpha_3$D and coupled binding and folding of TAD and NCBD. In three-color FRET, collecting single-molecule fluorescence trajectories at a high photon count rate is challenging because of various photophysical problems including frequent photoblinking, rapid photobleaching, and

relatively low quantum yield (i.e., low brightness) of the third fluorophore. To overcome these problems, we used Alexa 488, Alexa 594, and CF680R with reasonably high quantum yields and a chemical cocktail that reduces photoblinking and photobleaching (see Methods). Instead of ALEX/PIE, we employed a single donor excitation scheme with a CW laser that yields higher photon count rates and reduced photobleaching compared to PIE (Supplementary Fig. 1). This scheme is also useful to reduce background due to direct excitation of acceptor 2 in the binding experiment. The method determines three FRET efficiencies by globally analyzing three different kinds of photon trajectories: three-color, donor and A1 (DA1), donor and A2 (DA2) (Fig. 1). From the analysis of DA1 and DA2 photons, the energy transfer efficiency from D to A1 ($E_1$) or from D to A2 ($E_2$) is determined, which is required for the calculation of the other two FRET efficiencies. Since the two-color trajectories are always present in the experiment due to incomplete labeling or photobleaching of one of the two acceptor dyes during the measurement, all necessary data can be collected in a single experiment.

For the accurate determination of the kinetics on the μs–ms time scale, it is necessary to include acceptor photoblinking in the kinetic model (Supplementary Fig. 3) because acceptor blinking increases the apparent transition rates and changes the acceptor fraction values ($\varepsilon$)[18,27] despite the low populations of <5% in the acceptor dark states (see Supplementary Tables 1 and 2 for the comparison of the results with and without acceptor blinking corrections). Including acceptor blinking in the model may give an impression that much more fitting parameters are required because of the additional $\varepsilon$'s needed for the acceptor dark states. However, $\varepsilon$ values of the dark states of A1 and A2 can be related to $\varepsilon$'s of DA2 and DA1

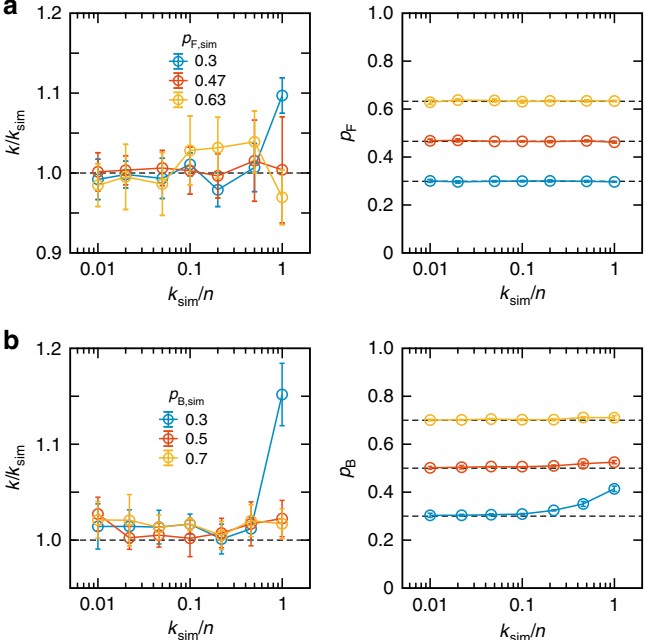

**Fig. 6 Determination of the relaxation rate and population for recolored photon trajectories.** See Supplementary Fig. 10 for the acceptor fractions. **a** $\alpha_3 D$ folding. Experimental photon trajectories collected at three different GdmCl concentrations (blue, 2 M, $p_F = 0.3$; red, 2.25 M, $p_F = 0.47$; yellow, 2.5 M, $p_F = 0.63$) were recolored using the corresponding maximum likelihood parameters except the relaxation rate ($k_F + k_U$, $k_{sim}$) that was varied in the simulation. **b** TAD/NCBD binding. Experimental photon trajectories at 0 mM NaCl were recolored using the extracted maximum likelihood parameters at 0 mM NaCl except the relaxation rate ($k_B + k_U$, $k_{sim}$) and bound fraction ($p_{B,sim}$, blue, 0.3; red, 0.5; yellow, 0.7) that were varied in the simulation. $n$ is the average photon count rate of the three-color segments, which are ~80 ms$^{-1}$ for $\alpha_3 D$ folding and 65 ms$^{-1}$ for TAD/NCBD binding. Horizontal dashed lines in $p_F$ and $p_B$ indicate the values in simulation. Error bars are standard deviations of the parameters obtained from five recolored data set. Source data are provided as a Source Data file.

trajectories, respectively (Supplementary Table 4), which results in no increase in the number of acceptor fraction parameters. Only four additional parameters are needed to account for acceptor blinking kinetics and acceptor bright state populations. Nevertheless, there are still a large number of fitting parameters in the global analysis (14 and 16 for two-state folding and two-state binding, respectively), and the likelihood optimization may look challenging. Indeed, it is time consuming to compute the likelihood function. We overcame this problem by developing a multi-thread calculation with CPU–GPU co-processing, which significantly reduces the computation time. (see Methods and Supplementary Fig. 2, the analysis codes are available at https://github.com/hoisunglab/FRET_3colorCW).

Using appropriate kinetic models, we showed that kinetic parameters (rates and relative population) can be determined robustly and the values are consistent with previous measurements. The simulation results show that it is possible to determine the transition rates close to the photon count rate (see Fig. 6) and detect an intermediate state (three-state model) with low population and short lifetime if it exists (Supplementary Fig. 11). We also compared the FRET efficiencies obtained from the three-color segments with those determined directly from the two-color segments after corrections for various experimental

factors. Compared to fractions of acceptor counts, accurate determination of FRET efficiencies is much more challenging due to the accumulation of the measurement errors of the parameters that are used for the conversion from the extracted acceptor fractions to the FRET efficiencies (Supplementary Fig. 8). More importantly, depending on the specific set of FRET efficiency values for a system, the uncertainty of the FRET efficiency determined from the three-color segments can be very large. For example, three-color $E_2^{3c}$ values determined both from 3c-CW using two-color $E_1^{2c}$ and from 3c-PIE[18] using $E_{12}^{2c}$ for $\alpha_3 D$ are inaccurate because $E_2$ is low. Since $E_2$ would be usually low due to the small spectral overlap between the donor and acceptor 2, this uncertainty can be a problem in many systems. In this case, $E_1^{3c}$ and $E_{12}^{3c}$ should be determined accurately using $E_2^{2c}$ found directly from the two-color parts of photon trajectories (Supplementary Fig. 8a). This result suggests that one should carefully choose a two-color $E$ value to be used for the calculation of three-color FRET efficiencies.

Finally, our development of fast three-color FRET experiment and analysis will be very useful in future structural and mechanistic studies of protein folding and binding. The mechanism of these processes can be understood best by following and visualizing structural evolution[9]. This is possible by probing protein folding transition path[9,11] or binding pathways of disordered proteins[30,31]. Folding and binding pathways are supposed to be heterogeneous, which can be characterized more effectively by using a multi-dimensional single-molecule tool such as three-color FRET than by a one-dimensional probe. These experiments will require even higher photon count rates for a better time resolution.

## Methods

**Protein expression and site-specific labeling.** After expression and purification, $\alpha_3 D$ was labeled with three fluorophores site-specifically[18]. Alexa 488 hydroxylamine (donor) and Alexa 594 maleimide (acceptor 1) were attached to 4-acetylphenylalanine[32,33] at the N-terminus and cysteine 33 of $\alpha_3 D$, respectively. Then, the C-terminal glycine residue of LPETG sequence was replaced by GGGC peptide using sortase-mediated ligation[34]. CF680R maleimide (acceptor 2) was subsequently attached to cysteine at the C-terminus.

After the expression and purification of TAD, Alexa 488 hydroxylamine and Alexa 594 maleimide were attached to the N-terminal 4-acetylphenylalanine and C-terminal cysteine residues, respectively[13]. CF680R maleimide was attached to cysteine 2066 (S2066C) of NCBD[13].

**Single-molecule experiments.** A confocal microscope system (MicroTime 200, PicoQuant) with an oil-immersion objective (UPLSAPO, NA 1.4, ×100, Olympus), a beamsplitter (z488/594rpc, Chroma Technology), and a 75 μm pinhole was used for single-molecule FRET experiments. Alexa 488 was excited by a 485 nm diode laser (LDH-D-C-485, PicoQuant) in the CW mode at 5.0 μW for $\alpha_3 D$ folding experiments and 3 μW for TAD/NCBD binding experiments, respectively. Fluorescence signal from three dyes was split into three photon counting avalanche photodiodes (SPCM-AQR-16, PerkinElmer Optoelectronics) using two dichroic beamsplitters (585DCXR and 670DCXR, Chroma Technology) and through bandpass filters (ET525/50 m for Alexa 488, ET645/75 m for Alexa 594, and ET705/72 m for CF680R, Chroma Technology).

Biotinylated $\alpha_3 D$ or TAD molecules were immobilized on a biotin-embedded, PEG-coated glass coverslip (Bio_01; Microsurfaces Inc.) via biotin (surface)-NeutrAvidin (Thermo Scientific)-biotin (protein) linkage. To minimize photoblinking and photobleaching of dyes, 100 mM β-mercaptoethanol (Sigma-Aldrich), 10 mM Cysteamine (Sigma-Aldrich)[35], 2 mM cyclooctatetraene (COT, Sigma-Aldrich), 2 mM 4-nitrobenzyl alcohol (NBA, Sigma-Aldrich), and 2 mM Trolox (Sigma-Aldrich)[36,37] were added into the solution. $\alpha_3 D$ folding experiments were performed in 50 mM HEPES, pH 7.4 (adjusted with NaOH) buffer with various GdmCl (Invitrogen) concentrations. TAD/NCBD binding experiments were performed in 10 mM Tris buffer (pH 7) with 0, 10, and 30 mM NaCl. All data were collected at room temperature (23°C).

**Single-molecule data analysis.** In the following sections, we describe the calculation of the likelihood functions of two- and three-color photon trajectories. The likelihood functions are optimized with respect to model parameters, which include the fractions of acceptor photon count rates (apparent FRET efficiencies in two-color FRET), rate coefficients, and relative populations in the states. The fractions of acceptor photon count rates are corrected for background, leak into other

detection channels (cross-talk), and direct excitation of the acceptors. The corrected fractions are then converted to three-color FRET efficiencies. The $\gamma$-factor (ratio of detection efficiencies and quantum yields) is taken into account during this conversion. We limit our discussion to the simplest two-state model for proteins. In the case of protein folding, these are the folded (F) and unfolded (U) states. In the case of fast binding, the two states are the bound (B) and unbound (U) states. The two-state model can be readily extended to more complex kinetic models. For example, the analysis code is applicable to the $N$-state linear model (see three-state model simulation and analysis in Supplementary Fig. 11). In addition to protein states, photophysical states of acceptors (i.e., bright and dark states) are incorporated in the kinetic models to account for acceptor blinking (See Supplementary Fig. 3).

**FRET efficiencies**. In the three-color experiment, the FRET efficiency is defined as the efficiency of the energy transfer between two fluorophores in the absence of a third fluorophore. For example, in Fig. 1a, $E_2$ is the transfer efficiency from the donor to acceptor 2 in the absence of acceptor 1. These FRET efficiencies are related to the distance between fluorophores as

$$E_1(r_1) = \frac{k_{\text{ET1}}}{k_{\text{D}} + k_{\text{ET1}}} = \frac{1}{1 + (r_1/R_1)^6}$$
$$E_2(r_2) = \frac{k_{\text{ET2}}}{k_{\text{D}} + k_{\text{ET2}}} = \frac{1}{1 + (r_2/R_2)^6} \qquad (1)$$
$$E_{12}(r_{12}) = \frac{k_{\text{ET12}}}{k_{\text{A1}} + k_{\text{ET12}}} = \frac{1}{1 + (r_{12}/R_{12})^6}.$$

Here, $k_{\text{D}}$ and $k_{\text{A1}}$ are the sums of the rates of the non-radiative and radiative decays of the donor and acceptor 1, respectively, in the absence of the energy transfer (Fig. 1a). $k_{\text{ET1}} = k_{\text{D}}(R_1/r_1)^6$, $k_{\text{ET2}} = k_{\text{D}}(R_2/r_2)^6$, and $k_{\text{ET12}} = k_{\text{A1}}(R_{12}/r_{12})^6$ are the rate constants of the energy transfer from D to A1, D to A2, and A1 to A2, respectively, and $R_1$, $R_2$, and $R_{12}$ are the corresponding Förster radii. The energy transfer rate constants depend on the distances $r_1$ (between D and A1), $r_2$ (between D and A2), and $r_{12}$ (between A1 and A2) (see Fig. 1a).

**Fractions of acceptor photon count rates**. The maximum likelihood method involves the fractions of acceptor photon count rates, instead of the FRET efficiencies above. The fractions of A1 and A2 photon count rates (denoted by $\varepsilon_1$ and $\varepsilon_2$) are defined as[18]

$$\varepsilon_1 = \frac{n_{\text{A1}}}{n_{\text{A1}} + n_{\text{A2}} + n_{\text{D}}}$$
$$\varepsilon_2 = \frac{n_{\text{A2}}}{n_{\text{A1}} + n_{\text{A2}} + n_{\text{D}}}. \qquad (2)$$

Here, $n_{\text{D}}$, $n_{\text{A1}}$, and $n_{\text{A2}}$, are the photon count rates (i.e., the numbers of photons per unit time) of the donor, acceptor 1, and acceptor 2, respectively.

For the two-color segments (DA1: donor and A1; DA2: donor and A2), the fractions of acceptor count rates are defined as[18]

$$\varepsilon^{\text{DA1}} = \frac{n_{\text{A1}}}{n_{\text{A1}} + n_{\text{D}}}$$
$$\varepsilon^{\text{DA2}} = \frac{n_{\text{A2}}}{n_{\text{A2}} + n_{\text{D}}}. \qquad (3)$$

Note that photons from two-color segments are also separated into three detection channels as those from three-color segments. In the analysis of DA1 segments, the photons detected in A2 channel result mostly from the leakage of A1 fluorescence. Therefore, A1 and A2 photons were combined and considered as A1 photons. Similarly, in the analysis of DA2 segments, D and A1 photons were combined and treated as donor photons.

**FRET efficiencies and fractions of acceptor photon counts**. The photon count rates detected after donor excitation can be expressed in terms of the rate constants[38] (see Fig. 1a)

$$n_{\text{D}} = \eta_{\text{D}} \phi_{\text{D}} k_{\text{D}}^{\text{ex}} \frac{k_{\text{D}}}{k_{\text{D}} + k_{\text{ET1}} + k_{\text{ET2}}}$$
$$n_{\text{A1}} = \eta_{\text{A1}} \phi_{\text{A1}} k_{\text{D}}^{\text{ex}} \frac{k_{\text{ET1}} k_{\text{A1}}}{(k_{\text{D}} + k_{\text{ET1}} + k_{\text{ET2}})(k_{\text{A1}} + k_{\text{ET12}})} \qquad (4)$$
$$n_{\text{A2}} = \eta_{\text{A2}} \phi_{\text{A2}} k_{\text{D}}^{\text{ex}} \left( \frac{k_{\text{ET2}}}{k_{\text{D}} + k_{\text{ET1}} + k_{\text{ET2}}} + \frac{k_{\text{ET1}} k_{\text{ET12}}}{(k_{\text{D}} + k_{\text{ET1}} + k_{\text{ET2}})(k_{\text{A1}} + k_{\text{ET12}})} \right).$$

Here, $k_{\text{D}}^{\text{ex}}$ is the donor excitation rate constant, $\eta_I$ and $\phi_I$ are the detection efficiency and quantum yield of fluorophore $I$ (=D, A1, and A2).
The count rates in Eq. (4) are related to the FRET efficiencies in Eq. (1)[38]

$$\frac{n_{\text{A1}}}{\gamma_1 n_{\text{D}}} = \frac{(1 - E_{12})E_1}{1 - E_1}$$
$$\frac{n_{\text{A2}}}{\gamma_2 n_{\text{D}}} = \frac{E_{12}E_1}{1 - E_1} + \frac{E_2}{1 - E_2}, \qquad (5)$$

where $\gamma_i = \phi_{\text{A}i}\eta_{\text{A}i}/\phi_{\text{D}}\eta_{\text{D}}$, $i = 1, 2$. These equations are used to find the FRET efficiencies in terms of the acceptor fractions determined in the maximum likelihood method. However, only two FRET efficiencies can be determined because there are only two equations.

If an alternating excitation scheme is used and $E_{12}$ is determined separately by additional excitation of A1, the other two FRET efficiencies, $E_1$ and $E_2$, can be determined from Eqs. (2) and (5) as[38]

$$E_1 = \left[ 1 + \varepsilon_1^{-1} \gamma_1 (1 - \varepsilon_1 - \varepsilon_2)(1 - E_{12}) \right]^{-1}$$
$$E_2 = \left[ 1 + (1 - \varepsilon_1 - \varepsilon_2) \left( \varepsilon_2 \gamma_2^{-1} - \varepsilon_1 \gamma_1^{-1} \frac{E_{12}}{1 - E_{12}} \right)^{-1} \right]^{-1}, \qquad (6)$$

Alternatively, if $E_1$ is known from two-color (DA1) segments, $E_2$ and $E_{12}$ are determined as

$$E_2 = \left[ 1 + \left( \frac{\varepsilon_1 \gamma_1^{-1} + \varepsilon_2 \gamma_2^{-1}}{1 - \varepsilon_1 - \varepsilon_2} - \frac{E_1}{1 - E_1} \right)^{-1} \right]^{-1}$$
$$E_{12} = 1 - \frac{\varepsilon_1 \gamma_1^{-1}(1 - E_1)}{(1 - \varepsilon_1 - \varepsilon_2)E_1}. \qquad (7)$$

Finally, when $E_2$ is known from two-color (DA2) segments, $E_1$ and $E_{12}$ are determined as

$$E_1 = \left[ 1 + \left( \frac{\varepsilon_1 \gamma_1^{-1} + \varepsilon_2 \gamma_2^{-1}}{1 - \varepsilon_1 - \varepsilon_2} - \frac{E_2}{1 - E_2} \right)^{-1} \right]^{-1}$$
$$E_{12} = \left[ 1 + \varepsilon_1 \gamma_1^{-1} \left( \varepsilon_2 \gamma_2^{-1} - (1 - \varepsilon_1 - \varepsilon_2) \frac{E_2}{1 - E_2} \right)^{-1} \right]^{-1}. \qquad (8)$$

In two-color FRET, the FRET efficiencies are found from the two-color acceptor fractions (see Eqs. (3) and (5) with $E_{12}$ set to 0)

$$E_i = \frac{\varepsilon_i}{\varepsilon_i + \gamma_i(1 - \varepsilon_i)}, \quad i = 1, 2. \qquad (9)$$

The fractions of acceptor photon count rates for the calculation of the FRET efficiencies above are corrected for background photons, detector cross-talk, and direct excitation of A1 and A2 (see below).

**Maximum likelihood method in the absence of acceptor blinking**. The likelihood function for a photon trajectory with records of photon colors and arrival times is[22]

$$L = \mathbf{1}^{\text{T}} \prod_{i=2}^{N} \left[ \mathbf{F}(c_i) \exp(\mathbf{K}(t_i - t_{i-1})) \right] \mathbf{F}(c_1) \mathbf{p}_{eq}, \qquad (10)$$

where $N$ is the number of photons in a trajectory, and $c_i$ and $t_i$ are the color and arrival time of the $i$th photon. $\mathbf{K}$ is the rate matrix, the photon color matrix $\mathbf{F}$ depends on the color $c$ of a photon as defined below for the two- and three-color cases. $\mathbf{1}^{\text{T}}$ is the unit row vector (T means transpose), and $\mathbf{p}_{eq}$ is the vector of equilibrium populations. The parameters were determined by maximizing the likelihood function calculated by diagonalizing $\mathbf{K}$[22]. Practically, the total log-likelihood function of all trajectories was calculated by summing individual log-likelihood functions in Eq. (10).

In the calculation of the likelihood function for a two-color segment with fluorescence of donor and one of the two acceptors in protein folding experiment, the photon color matrix is $\mathbf{F}(\text{acceptor}) = \mathbf{E}$ and $\mathbf{F}(\text{donor}) = \mathbf{I} - \mathbf{E}$, where $\mathbf{E}$ is a diagonal matrix with the fractions of acceptor photon count rates of the individual states on the diagonal and $\mathbf{I}$ is the identity matrix. For the two-state model, the matrix of fractions of acceptor photons, the rate matrix, and the vector of the equilibrium populations are given by

$$\mathbf{E} = \begin{pmatrix} \varepsilon_{\text{F}} & 0 \\ 0 & \varepsilon_{\text{U}} \end{pmatrix},$$
$$\mathbf{K} = \begin{pmatrix} -k_{\text{U}} & k_{\text{F}} \\ k_{\text{U}} & -k_{\text{F}} \end{pmatrix}, \quad \mathbf{p}_{eq} = \begin{pmatrix} p_{\text{F}} \\ 1 - p_{\text{F}} \end{pmatrix}, \qquad (11)$$

where $\varepsilon_{\text{F}}$ and $\varepsilon_{\text{U}}$ are the fractions of acceptor photons of the folded and unfolded states defined in Eq. (3), and $p_{\text{F}} = k_{\text{F}}/(k_{\text{F}} + k_{\text{U}})$ is the equilibrium population of the folded state. $k_{\text{F}}$ and $k_{\text{U}}$ are the folding and unfolding rate coefficients.

For a three-color segment, $\mathbf{F}(\text{acceptor 1}) = \mathbf{E}_1$, $\mathbf{F}(\text{acceptor 2}) = \mathbf{E}_2$, and $\mathbf{F}(\text{donor}) = \mathbf{I} - \mathbf{E}_1 - \mathbf{E}_2$, where $\mathbf{E}_1$ and $\mathbf{E}_2$ are the diagonal matrices with the fractions of acceptor 1 (A1) and acceptor 2 (A2) photons defined in Eq. (2).

In the case of protein folding experiment, all three dyes are attached to the same molecule. Two- and three-color segments are independent, so the corresponding likelihoods can be calculated separately. The total likelihood is the product of individual ones with different dye combinations (i.e., $L = L^{(3c)}L^{(DA1)}L^{(DA2)}$). (If photobleaching or slow blinking of one acceptor occurs, the three-color segment and two-color segment before and after photobleaching are not independent, but this effect would be negligible in the parameter determination.) This likelihood function is optimized with respect to

the relaxation rate, $k$ $(= k_F + k_U)$, equilibrium population of the folded state, $p_F = k_F/k$, three-color acceptor fractions, $\varepsilon_{1F}, \varepsilon_{2F}, \varepsilon_{1U}, \varepsilon_{2U}$, and two-color acceptor fractions, $\varepsilon_F^{DA1}, \varepsilon_U^{DA1}, \varepsilon_F^{DA2}, \varepsilon_U^{DA2}$, in the folded and unfolded states.

In the case of fast binding, because of incomplete labeling and photobleaching of A2, binding to both labeled and unlabeled binding partners can take place (see Supplementary Fig. 3d). Therefore, the likelihood function for three-color part and two-color part with the donor and A1 (DA1) must be calculated together with a single rate matrix. The matrices for 3-color/DA1 segments are

$$
\mathbf{E}_1 = \begin{pmatrix} \varepsilon_{1Bb} & 0 & 0 \\ 0 & \varepsilon_{1U} & 0 \\ 0 & 0 & \varepsilon_{1Bd} \end{pmatrix}, \quad \mathbf{E}_2 = \begin{pmatrix} \varepsilon_{2Bb} & 0 & 0 \\ 0 & \varepsilon_{2U} & 0 \\ 0 & 0 & \varepsilon_{2Bd} \end{pmatrix},
$$
$$
\mathbf{K} = \begin{pmatrix} -k_U - k'_{d2} & \varphi k_B & k'_{b2} \\ k_U & -k_B & k_U \\ k'_{d2} & (1-\varphi)k_B & -k_U - k'_{b2} \end{pmatrix}, \quad \mathbf{p}_{eq} = \begin{pmatrix} \varphi p_B \\ 1 - p_B \\ (1-\varphi)p_B \end{pmatrix}. \tag{12}
$$

Here, $\varepsilon_{1Bb}$ and $\varepsilon_{1Bd}$ are the fractions of acceptor $I$ ($I = 1, 2$) photons of the bound state with an active and inactive (or unlabeled) A2, respectively, and $\varepsilon_{1U}$ is the fractions of acceptor photons of the unbound state. There is only one unbound state because A2 is attached to NCBD and not present in unbound TAD. $\varphi$ is the fraction of the molecules with active A2. Since photobleaching and photoblinking of A2 occur on a timescale comparable or slower than the binding rate, these dynamics are also incorporated by adding an effective fraction of the A2 bright state and rates of bleaching/blinking of A2 as $\varphi' = k'_{b2}/(k_{b2} + k'_{d2})$, where $k'_{b2}$ and $k'_{d2}$ are the rate coefficients of photobleaching or slow blinking of A2. Since $\varphi' < \varphi$ due to the irreversible photobleaching, the detailed balance does not hold. There are six acceptor fractions in the likelihood in Eq. (12). These include A2 fractions in the A2 dark state, $\varepsilon_{2Bd}$ and $\varepsilon_{2U}$, which are mostly due to leak and background photons. To relate them, we use the fact that the ratio of the count rate of A2 to that of A1 is approximately the same for both bound and unbound states (i.e., $\varepsilon_{2Bd}/\varepsilon_{1Bd} = \varepsilon_{2U}/\varepsilon_{1U}$) because A2 photons will consist of the leak of A1 photons into the A2 channel when background photons are excluded. Using this relationship, the number of independent acceptor fractions reduces to 5.

The likelihood function of a DA2 segment is calculated using the rate matrix and equilibrium population in Eq. (12), with $\mathbf{E} = Diag(\varepsilon_{Bb}^{DA2}, \varepsilon_U^{DA2}, \varepsilon_{Bd}^{DA2})$. Here, $\varepsilon_{Bd}^{DA2} = \varepsilon_U^{DA2}$ because the bound state with inactive A2 is indistinguishable from the unbound state (i.e. donor only). The total likelihood is the product of the likelihoods of 3color/DA1 and DA2 segments (i.e., $L = L^{(3c/DA1)}L^{(DA2)}$). This is optimized with respect to 12 fitting parameters ($\varepsilon_{1Bb}$, $\varepsilon_{1Bd}$, $\varepsilon_{2Bd}$, $\varepsilon_{1U}$, $\varepsilon_{Bb}^{DA2}$, $\varepsilon_U^{DA2}$, $k_B$, $k_U$, $\varphi$, $k'_{b2}$, and $k'_{d2}$). Fractions $\varepsilon_{1Bb}$, $\varepsilon_{2Bb}$, are used to find three-color FRET efficiencies $E_{1B}^{3c}$ and $E_{2B}^{3c}$ in the bound state. Fractions $\varepsilon_{1Bd}$ and $\varepsilon_{1U}$ are used to find two-color FRET efficiencies $E_{1B}^{2c}$ and $E_{1U}^{2c}$ in the bound and unbound states, respectively. Finally, $\varepsilon_{Bb}^{DA2}$ from DA2 segments is used to find two-color FRET efficiency $E_{2B}^{2c}$ in the bound state (see Fig. 1d).

**Maximum likelihood method in the presence of blinking.** In the two-state folding analysis of two-color segments with acceptor blinking, the matrices are

$$
\mathbf{E} = \begin{pmatrix} \varepsilon_F & 0 & 0 & 0 \\ 0 & \varepsilon_U & 0 & 0 \\ 0 & 0 & \varepsilon_d & 0 \\ 0 & 0 & 0 & \varepsilon_d \end{pmatrix},
$$
$$
\mathbf{K} = \begin{pmatrix} -k_U - k_d & k_F & k_b & 0 \\ k_U & -k_F - k_d & 0 & k_b \\ k_d & 0 & -k_U - k_b & k_F \\ 0 & k_d & k_U & -k_F - k_b \end{pmatrix}, \quad \mathbf{p}_{eq} = \begin{pmatrix} p_F p_b \\ (1-p_F)p_b \\ p_F(1-p_b) \\ (1-p_F)(1-p_b) \end{pmatrix}, \tag{13}
$$

where $\varepsilon_d$ is the fraction of the acceptor photons in the dark state and $p_b = k_b/(k_b + k_d)$ is the equilibrium population of the acceptor bright state. $k_b$ and $k_d$ are the rate coefficients from the dark to the bright states and vice versa. We assumed that $k_d$ is proportional to the photon count rate because $k_d$ increases linearly with the time spent in the excited state, while $k_b$ is independent of the photon count rate. Therefore, $k_d = k_d^0(n/n_0)$, where $n$ is the average photon count rate of each photon trajectory and $k_d^0$ is the rate coefficient at the reference photon count rate ($n_0 = 100$ ms$^{-1}$).

In the two-state folding analysis of three-color segments with acceptor blinking (see Supplementary Fig. 3c),

$$
\mathbf{E}_1 = Diag(\{\varepsilon_{1Sjk}\})
$$
$$
\mathbf{E}_2 = Diag(\{\varepsilon_{2Sjk}\})
$$
$$
\mathbf{p}_{eq} = (\{p_{Sjk}\})^T, \quad S = F, U, \quad j, k = b, d
$$
$$
\mathbf{K} = \begin{pmatrix} -k_U & k_F & k_{b2} & 0 & k_{b1} & 0 & 0 & 0 \\ -k_{d1} - k_{d2} & & & & & & & \\ k_U & -k_F & 0 & k_{b2} & 0 & k_{b1} & 0 & 0 \\ & -k_{d1} - k_{d2} & & & & & & \\ k_{d2} & 0 & -k_U & k_F & 0 & 0 & k_{b1} & 0 \\ & & -k_{d1} - k_{b2} & & & & & \\ 0 & k_{d2} & k_U & -k_F & 0 & 0 & 0 & k_{b1} \\ & & & -k_{d1} - k_{b2} & & & & \\ k_{d1} & 0 & 0 & 0 & -k_U & k_F & k_{b2} & 0 \\ & & & & -k_{b1} - k_{d2} & & & \\ 0 & k_{d1} & 0 & 0 & k_U & -k_F & 0 & k_{b2} \\ & & & & & -k_{b1} - k_{d2} & & \\ 0 & 0 & k_{d1} & 0 & k_{d2} & 0 & -k_U & k_F \\ & & & & & & -k_{b1} - k_{b2} & \\ 0 & 0 & 0 & k_{d1} & 0 & k_{d2} & k_U & -k_F \\ & & & & & & & -k_{b1} - k_{b2} \end{pmatrix}, \tag{14}
$$

Here, $\varepsilon_{ISjk}$ is the fraction of acceptor $I$ ($= 1, 2$) photons of state $S$ ($= F, U$) with the bright (b) and dark (d) states of acceptor 1 ($j$) and acceptor 2 ($k$). For example, $\varepsilon_{1Fbd}$ is the fraction of A1 in the folded state with A1 in the bright and A2 in the dark states. $p_{Sjk} = p_S \times p_{j1} \times p_{k2}$, and $p_{bI} = k_{bI}/(k_{bI} + k_{dI})$ is the equilibrium population of the bright state of acceptor $I$. $k_{bI}$ and $k_{dI}$ are the rate coefficients from the dark to the bright states and vice versa of acceptor $I$.

In the global analysis of two- and three-color segments, the number of independent acceptor fraction parameters can be reduced by using the relationships in Supplementary Table 4. Overall, there are 14 fitting parameters including 8 acceptor fractions ($\varepsilon_F^{DA1}, \varepsilon_U^{DA1}, \varepsilon_F^{DA2}$, and $\varepsilon_U^{DA2}$ for 2-color DA1 and DA2 segments and $\varepsilon_{1F}, \varepsilon_{1U}, \varepsilon_{2F}, \varepsilon_{2U}$ for 3-color segments), $k_F$ and $k_U$, and four rate coefficients for A1 and A2 blinking. Fixed parameters are listed in Supplementary Table 4.

In the analysis of three-color/DA1 segments of two-state binding with acceptor blinking (see Supplementary Fig. 3f),

$$
\mathbf{E}_1 = Diag(\varepsilon_{1Bbb}, \varepsilon_{1Ubb}, \varepsilon_{1Bbd}, \varepsilon_{1Bdb}, \varepsilon_{1Udb}, \varepsilon_{1Bdd}, \varepsilon_{1Bdd})
$$
$$
\mathbf{E}_2 = Diag(\varepsilon_{2bbb}, \varepsilon_{2Ubb}, \varepsilon_{2Bbd}, \varepsilon_{2Bdb}, \varepsilon_{2Udb}, \varepsilon_{2Bdd}, \varepsilon_{2Bdd})
$$
$$
\mathbf{P}_{eq} = (p_B p_{b1} p_{b2} \varphi, p_U p_{b1}, p_B p_{b1}(1-\varphi), p_B p_{b1} p_{d2} \varphi, p_B p_{d1} p_{b2} \varphi, p_U p_{d1}, p_B p_{d1}(1-\varphi), p_B p_{d1} p_{d2} \varphi)^T
$$
$$
\mathbf{K} = \begin{pmatrix}
-k_U - k_{d1} & \varphi p_{b2} k_B & k'_{b2} & k_{b2} & k_{b1} & 0 & 0 & 0 \\
-k_{d2} - k'_{d2} & & & & & & & \\
k_U & -k_B - k_{d1} & k_U & k_U & 0 & k_{b1} & 0 & 0 \\
k'_{d2} & (1-\varphi)k_B & -k_U - k_{d1} & 0 & 0 & 0 & k_{b1} & 0 \\
& & -k'_{b2} & & & & & \\
k_{d2} & \varphi p_{d2} k_B & 0 & -k_U - k_{d1} & 0 & 0 & 0 & k_{b1} \\
& & & -k_{b2} & & & & \\
k_{d1} & 0 & 0 & 0 & -k_U - k_{b1} & \varphi p_{b2} k_B & k'_{b2} & k_{b2} \\
& & & & -k_{d2} - k'_{d2} & & & \\
0 & k_{d1} & 0 & 0 & k_U & -k_B - k_{b1} & k_U & k_U \\
0 & 0 & k_{d1} & 0 & k'_{d2} & (1-\varphi)k_B & -k_U - k_{b1} & 0 \\
& & & & & & -k'_{b2} & \\
0 & 0 & 0 & k_{d1} & k_{d2} & \varphi p_{d2} k_B & 0 & -k_U - k_{b1} \\
& & & & & & & -k_{b2}
\end{pmatrix}, \tag{15}
$$

The definition of the matrix elements of $\mathbf{E}_1$ and $\mathbf{E}_2$ are the same as those defined in Eq. (14) except that subscripts B and U stand for the bound and unbound states, respectively. As indicated in the vector of equilibrium population, the first three states are the same as the three states in Eq. (12) with A1 in the bright state. The forth state is A1 bright and A2 dark state to account for fast A2 blinking. The last four states are a replica of the first four states with A1 in the dark states. Note that a subscript "b" is used for the state of A2 in the unbound state acceptor fractions, $\varepsilon_{Ubb}$ and $\varepsilon_{Udb}$, for the consistency of the notations, but this does not represent A2 bright state because A2 is not present in the unbound state. In addition, similar to the situation without considering acceptor blinking, one fitting parameter can be reduced using the relationship for the unbound state, $\varepsilon_{2Bbd}/\varepsilon_{1Bbd} = \varepsilon_{2Ubb}/\varepsilon_{1Ubb}$.

The likelihood function of DA2 segments can be calculated with a four-state model: three states in Eq. (12) and an additional A2 dark state. The kinetic scheme is the same as the bright (or dark) half of A1 state in Supplementary Fig. 3f. The matrix elements are

$$
\mathbf{E} = Diag(\varepsilon_B^{DA2}, \varepsilon_U^{DA2}, \varepsilon_{d2}, \varepsilon_{d2})
$$
$$
\mathbf{P}_{eq} = (p_B p_{b2} \varphi, p_U, p_B(1-\varphi), p_B p_{d2} \varphi)^T
$$
$$
\mathbf{K} = \begin{pmatrix}
-k_U & \varphi p_{b2} k_B & k'_{b2} & k_{b2} \\
-k_{d2} - k'_{d2} & & & \\
k_U & -k_B & k_U & k_U \\
k'_{d2} & (1-\varphi)k_B & -k_U - k'_{b2} & 0 \\
k_{d2} & \varphi p_{d2} k_B & 0 & -k_U - k_{b2}
\end{pmatrix}. \tag{16}
$$

Here, the acceptor fraction of A2 dark states (third and fourth states in Eq. (16)) are the same as that of the unbound state as $\varepsilon_{d2} = \varepsilon_U^{DA2}$, which is equal to the A2 fraction of donor-only segments. There are 16 fitting parameters in this model, $L = L^{(3c/DA1)}L^{(DA2)}$, which includes seven parameters in $\mathbf{E}$ matrices, $k_B$ and $k_U$, and

four rate coefficients for blinking of A1 and A2, labeling efficiency of A2 ($\varphi$), and two rate coefficients for photobleaching (and slow blinking) of A2.

**Correction of the fractions of photon counts.** To find FRET efficiencies, the values of the fractions of acceptor photons determined from the maximum likelihood analysis were corrected for background, leak into other channels, and direct acceptor excitation[18,38,39] as described in the following sections.

**Corrections for background.** To correct for background, the photon count rates are subtracted by the background photon count rates of the correspondent detection channels. As a result, the fractions of acceptor photons defined in Eqs. (2) and (3) are corrected for background as[39]

$$\varepsilon^c = \frac{\varepsilon n - b_A}{n - b_A - b_D} \quad \text{(2 color).}$$

$$\varepsilon_1^c = \frac{\varepsilon_1 n - b_{A1}}{n - b_D - b_{A1} - b_{A2}}, \quad \varepsilon_2^c = \frac{\varepsilon_2 n - b_{A2}}{n - b_D - b_{A1} - b_{A2}}, \quad \text{(3 color).} \tag{17}$$

Here, $\varepsilon$'s are the uncorrected values obtained from the maximum likelihood analysis above, $n$ is the average total photon count rate including background photons, and $b_D$, $b_{A1}$, and $b_{A2}$ are the average background count rates in D, A1, and A2 channels, respectively. Average background count rates were obtained from the segments after all dyes are photobleached. In the binding experiment, $b_{A2}$ includes A2 photons emitted from NCBD in solution by direct A2 excitation.

**Corrections for donor and A1 leak.** Some donor photons can be detected as acceptor photons (donor leak to the acceptor channels). In two-color segments, the photon count rates with donor leak are $n_A = n_A^c + l n_D^c$, $n_D = (1-l) n_D^c$, where $n_A^c$ and $n_D^c$ denote the count rates without donor leak, $l$ is the fraction of donor photons detected in the acceptor channel. In three-color segments, the photon count rates are $n_{A1} = (1 - l_{12}) n_{A1}^c + l_1 n_D^c$, $n_{A2} = l_{12} n_{A1}^c + n_{A2}^c + l_2 n_D^c$, $n_D = (1 - l_1 - l_2) n_D^c$. These count rates are used in the acceptor fractions with ($\varepsilon$) and without ($\varepsilon^c$) donor leak. Thus, for the two-color segments, the donor leak is corrected as[18]

$$\varepsilon^c = \frac{\varepsilon - l}{1 - l}, \tag{18}$$

where $\varepsilon$ is the value corrected for background photons in Eq. (17) and $l$ ($= n_A^0/(n_A^0 + n_D^0)$) is the average value of the leak of donor photons into the acceptor channel, which can be determined using $n_A^0$ and $n_D^0$, the background-corrected mean photon count rates in the acceptor and donor channels of donor-only segments.

For the three-color segments, the leak of the donor photons into the two acceptor channels can be corrected as[18]

$$\varepsilon_1^c = \frac{\varepsilon_1(1 - l_2) - (1 - \varepsilon_2)l_1}{(1 - l_1 - l_2)(1 - l_{12})}$$

$$\varepsilon_2^c = \frac{\varepsilon_1(l_2 - l_{12}) + \varepsilon_2(1 - l_1 - l_{12}) + (l_1 + l_2)l_{12} - l_2}{(1 - l_1 - l_2)(1 - l_{12})}. \tag{19}$$

Here, $l_1 = n_{A1}^0/(n_{A2}^0 + n_{A1}^0 + n_D^0)$ and $l_2 = n_{A2}^0/(n_{A2}^0 + n_{A1}^0 + n_D^0)$. $n_{A2}^0$, $n_{A1}^0$, and $n_D^0$ are the background-corrected mean photon count rates in A2, A1, and D channels of donor-only segments. $l_{12} = (n\varepsilon_2 - b_{A2})/(n\varepsilon_1 + n\varepsilon_2 - b_{A1} - b_{A2})$, where $\varepsilon_1$ and $\varepsilon_2$ are the acceptor fractions obtained from DA1 segments in the folding experiment and fractions of photons detected in A1 and A2 channels for the unbound state in the binding analysis. The experimental values are $l_1 = 0.051$, $l_2 = 0.001$, and $l_{12} = 0.23$ for $\alpha_3$D folding and $l_1 = 0.046$, $l_2 = 0.007$, and $l_{12} = 0.17$ for TAD/NCBD binding.

**Correction for direct acceptor excitation.** A small fraction of acceptor photons results from direct excitation of the acceptor by a laser instead of the energy transfer from the donor. The rate of the acceptor excitation is $k_D^{ex} f^{dir} (f^{dir} < 1)$. (Note that the definition of $f^{dir}$ differs from those in refs. [18,38] by a multiplication factor $\gamma$.) Therefore, in the two-color segments, the acceptor photon count rate with direct acceptor excitation is $n_A = n_A^c + n_A^{dir}$, where $n_A^{dir} = \eta_A \phi_A k_D^{ex} f^{dir} = (n_A^c + \gamma n_D^c) f^{dir}$. The donor count rate does not change. This count rate is used in Eq. (3) for the fraction of acceptor photons with ($\varepsilon$) and without ($\varepsilon^c$) direct acceptor excitation. The resulting correction for the direct excitation in the two-color analysis is

$$\varepsilon^c = \frac{\varepsilon - f^{dir} \gamma (1 - \varepsilon)}{1 + f^{dir}(1 - \gamma)(1 - \varepsilon)}, \tag{20}$$

Here, $\varepsilon$ is the background and donor-leak corrected acceptor fraction in Eq. (18). $f^{dir} = n_A^{dir}/(n_A + \gamma n_D)$ can be experimentally measured by the ratio of the photon count rates before ($n_A$ and $n_D$) and after ($n_A^{dir}$) donor bleaching but prior to acceptor bleaching.

In the three-color segments, the acceptor photon count rates with direct acceptor excitation are $n_{A1} = n_{A1}^c + n_{A1}^{dir}$ and $n_{A2} = n_{A2}^c + n_{A2}^{dir}$, and the donor count rate does not change. The rates of A1 and A2 direct excitation are $k_D^{ex} f_1^{dir}$ and $k_D^{ex} f_2^{dir}$, respectively. Using this in Eq. (2), we find the corrected fraction of acceptor

photons ($i = 1, 2$)

$$\varepsilon_i^c = \frac{n_{Ai}^c}{n_{A1}^c + n_{A2}^c + n_D} = \frac{\varepsilon_i - x_i}{1 - x_1 - x_2}. \tag{21}$$

where $x_i = (1 - \varepsilon_1 - \varepsilon_2) n_{Ai}^{dir}/n_D$. To specify $n_{Ai}^{dir}/n_D$, we note that a fraction of the A1 excitation transfers to A2, which makes the correction more complicated than in the two-color case. The count rate of A1 due to direct excitation is $n_{A1}^{dir} = \eta_{A1}\phi_{A1} k_D^{ex} f_1^{dir} (1 - E_{12})$. The change in the A2 count rate is (1) due to the direct excitation of A1 and subsequent energy transfer to A2 and (2) due to the direct excitation of A2, i.e., $n_{A2}^{dir} = \eta_{A2}\phi_{A2} k_D^{ex} (f_1^{dir} E_{12} + f_2^{dir})$. Then, using $\eta_{Ai}\phi_{Ai} = \gamma_i \eta_D \phi_D$, $\eta_D \phi_D k_D^{ex} = n_{A1}^c \gamma_1^{-1} + n_{A2}^c \gamma_2^{-1} + n_D$ (as follows from Eq. (4)), and Eq. (5) for $n_{Ai}^c/\gamma_i n_D$, we find $x_i$. As a result, we have the following three-color acceptor fractions corrected for direct acceptor excitation

$$\varepsilon_1^c = \frac{\varepsilon_1 - f_1^{dir}\gamma_1(1 - E_{12})Z}{1 - (f_1^{dir}\gamma_1 + f_2^{dir}\gamma_2 + f_1^{dir}(\gamma_2 - \gamma_1)E_{12})Z}$$

$$\varepsilon_2^c = \frac{\varepsilon_2 - (f_2^{dir} + f_1^{dir}E_{12})\gamma_2 Z}{1 - (f_1^{dir}\gamma_1 + f_2^{dir}\gamma_2 + f_1^{dir}(\gamma_2 - \gamma_1)E_{12})Z}$$

$$Z = (1 - \varepsilon_1 - \varepsilon_2)\left(1 + \frac{E_1}{1 - E_1} + \frac{E_2}{1 - E_2}\right). \tag{22}$$

Here, $\varepsilon_1$ and $\varepsilon_2$ are the background and donor-leak corrected fractions (Eqs. (17) and (19)), $E_1$, $E_2$, and $E_{12}$ are the true FRET efficiencies, which are found from two-color segments and/or using Eqs. (6)–(8) iteratively. For example, when $E_2^{2c}$ is known (i.e., determined from DA2 segments), $E_1^{3c}$ and $E_{12}^{3c}$ are found iteratively using Eqs. (8) and (22) with uncorrected $\varepsilon_1$ and $\varepsilon_2$ as initial values. Since $f_1^{dir}$ and $f_2^{dir}$ are small, the values converge after several iterations.

Experimentally, the $\gamma$-factors were determined by comparing the count rates before and after acceptor photobleaching. $\gamma_1 = (\eta_{A1}\phi_{A1})/(\eta_D\phi_D)$ is 1.51 and $\gamma_2 = (\eta_{A2}\phi_{A2})/(\eta_D\phi_D)$ is 0.75 for $\alpha_3$D folding, and $\gamma_1 = 0.83$ and $\gamma_2 = 0.49$ for TAD/NCBD binding. $f_1^{dir}$ and $f_2^{dir}$ are 0.067 and 0.072, respectively. Since the fluorescence signal by direct excitation of A2 of unbound NCBD in solution is treated as background in A2 channel, we set $f_2^{dir} = 0$ for the unbound state (TAD only) in the binding experiment.

**Gaussian chain model.** When the inter-dye distances fluctuate, the FRET efficiencies determined from the three-color segments differ from those determined from the two-color segments[18]. The two-color FRET efficiency is the average with respect to the fluctuation of only one donor–acceptor distance. In three-color FRET, fluctuations of all three distances influence the measured acceptor fractions and the FRET efficiencies calculated using Eqs. (6)–(8). Therefore, it is natural that the FRET efficiencies obtained from the two-color and from the three-color segments are different in the case of fluctuating distances. This effect can be evaluated by calculating the three-color FRET efficiencies using the Gaussian chain model for the unfolded state of the protein. The parameters of the model are extracted from the two-color FRET efficiencies.

The Gaussian chain model involves only the mean-squared distance $\langle r_i^2 \rangle$ as a parameter. In $\alpha_3$D construct, the two vectors, $\mathbf{r}_1$ connecting D and A1 and $\mathbf{r}_{12}$ connecting A1 and A2 (Fig. 1a), are assumed to be independent and have a Gaussian distribution, so $\mathbf{r}_2 = \mathbf{r}_1 + \mathbf{r}_{12}$ has also the Gaussian distribution with $\langle r_2^2 \rangle = \langle r_1^2 \rangle + \langle r_{12}^2 \rangle$. Thus, the inter-dye distance distributions are

$$p_i(r_i) = \left(2\pi\langle r_i^2 \rangle/3\right)^{-3/2} \exp\left(-\frac{3r_i^2}{2\langle r_i^2 \rangle}\right), \qquad i = 1, 2, 12. \tag{23}$$

These distributions are normalized as $\int p_i(r_i) d\mathbf{r}_i = \int_0^\infty p_i(r_i) 4\pi r_i^2 dr_i = 1$.

The FRET efficiency $E_i$ determined from the two-color photon trajectories is averaged as

$$\langle E_i \rangle = \int_0^\infty \frac{p_i(r_i) 4\pi r_i^2}{1 + r_i^6/R_i^6} dr_i. \tag{24}$$

$\langle r_1^2 \rangle$ and $\langle r_2^2 \rangle$ are obtained by fitting the experimental FRET efficiencies $E_1^{2c}$ and $E_2^{2c}$ from the two-color segments to Eq. (24) and $\langle r_{12}^2 \rangle$ is found as $\langle r_2^2 \rangle - \langle r_1^2 \rangle$. When the alternating excitation is used, $\langle r_{12}^2 \rangle$ can also be determined from the two-color FRET efficiency $E_{12}$ by A1 excitation.

The three-color FRET efficiencies due to the Gaussian chain model are found using Eqs. (6)–(8) with the fractions of acceptor photons replaced by those with the averaged photon counts[18], $\varepsilon_1 \to \langle n_{A1}\rangle/(\langle n_{A1}\rangle + \langle n_{A2}\rangle + \langle n_D\rangle)$ and $\varepsilon_2 \to \langle n_{A2}\rangle/(\langle n_{A1}\rangle + \langle n_{A2}\rangle + \langle n_D\rangle)$. To find $\langle n_{A1}\rangle$, $\langle n_{A2}\rangle$, and $\langle n_D\rangle$, we use $k_{ET1} = k_D(R_1/r_1)^6$, $k_{ET2} = k_D(R_2/r_2)^6$, and $k_{ET12} = k_{A1}(R_{12}/r_{12})^6$ in Eq. (4) and present the photon count rates as

$$n_D(r_1, r_2) = c^{Dex}\left(1 + \frac{R_1^6}{r_1^6} + \frac{R_2^6}{r_2^6}\right)^{-1}$$

$$n_{A1}(r_1, r_2, r_{12}) = \gamma_1 n_D(r_1, r_2) \frac{R_1^6}{r_1^6} \frac{r_{12}^6}{r_{12}^6 + R_{12}^6}$$

$$n_{A2}(r_1, r_2, r_{12}) = \gamma_2 n_D(r_1, r_2)\left(\frac{R_1^6}{r_1^6} \frac{R_{12}^6}{r_{12}^6 + R_{12}^6} + \frac{R_2^6}{r_2^6}\right), \tag{25}$$

where $c^{Dex} = \eta_D \phi_D k_D^{ex}$ is a constant, which cancels when calculating the acceptor fractions.

The acceptor count rates are averaged as ($i = 1, 2$)

$$\langle n_{Ai} \rangle = \int n_{Ai}(r_1, r_2, r_{12}) p_1(r_1) p_{12}(r_{12}) \, d\mathbf{r}_1 d\mathbf{r}_{12}$$
$$= 8\pi^2 \int_0^\infty dr_1 \int_0^\infty dr_{12} \int_0^\pi d\theta \, n_{Ai}(r_1, r_2, r_{12}) p_1(r_1) p_{12}(r_{12}) r_1^2 r_{12}^2 \sin \theta . \quad (26)$$

where $r_2$ in $n_{Ai}(r_1, r_2, r_{12})$ is $r_2 = \sqrt{r_1^2 + r_{12}^2 - 2r_1 r_{12} \cos \theta}$ (see Fig. 1a).

The donor count rates depend only on $r_1$ and $r_2$, so the averaging is simplified as

$$\langle n_D \rangle = \int n_D(r_1, r_2) p_1(r_1) p_{12}(|\mathbf{r}_2 - \mathbf{r}_1|) \, d\mathbf{r}_1 d\mathbf{r}_2$$
$$= \frac{8\pi^2}{3} \langle r_{12}^2 \rangle \int_0^\infty dr_1 \int_0^\infty dr_2 \, n_D(r_1, r_2) p_1(r_1) [p_{12}(r_1 - r_2) - p_{12}(r_1 + r_2)] r_1 r_2 . \quad (27)$$

The averaged count rates in Eqs. (26) and (27) are used in Eq. (2) and then in Eqs. (6)–(8). $E_1$ ($= E_1^{2c}$), $E_2$ ($= E_2^{2c}$), and $E_{12}$ ($= E_{12}^{2c}$) on the right-hand side of Eqs. (6)–(8) are determined from the two-color segments. In this way, we get the three-color FRET efficiencies on the left-hand side obtained using the Gaussian model with the parameters from the two-color FRET efficiencies. The two-color and calculated three-color FRET efficiencies are compared in Supplementary Fig. 9 for various sets of two independent parameters, $\langle r_1^2 \rangle$ and $\langle r_{12}^2 \rangle$. The Förster radii, $R_1 = 5.4$ nm, $R_2 = 4.3$ nm and $R_{12} = 7.0$ nm used in the calculation were obtained from the spectral overlap between measured dye spectra. The calculation shows that the difference between the three-color and two-color FRET efficiencies due to fluctuations of the distances in the unfolded state can be significant. However, for the experimental parameters of $\alpha_3$D folding, the deviation does not exceed 2.5%.

**State assignment at the single photon level**. The transitions between states can be identified at the single photon level using the parameters extracted using the maximum likelihood method and the Viterbi algorithm[40,41] adapted to photon trajectory analysis[18,25] (see Supplementary Fig. 5). The sequence of states obtained from the photon trajectory is overlaid in the binned fluorescence trajectory as an example, as indicated in the color bar above the trajectory in the upper panel of Fig. 3a.

**CPU–GPU co-parallelization for the likelihood calculation**. To find the most likely parameters we used a derivative-free optimization method (Nelder–Mead). The optimization process involves iterative evaluation of the likelihood that requires a large number of matrix–vector multiplications which is of the same order of magnitude as the number of photons analyzed. In the analysis of three-color data, it takes a long time to maximize the likelihood due to the large number of parameters and the large matrix size. To accelerate the calculation, we parallelized the likelihood calculation using both GPU and CPU. The basic idea is to parallelize the evaluation of likelihood values of the segments ($N_S$) of photon trajectories (Supplementary Fig. 2a). First, the number of segments to be calculated by GPU ($N_G$) is determined based on a pre-performed benchmark result, which depends on the specifications of processors. During the optimization process, the likelihood values for segment 1 to $N_G$ are calculated on GPU and the values for the rest of segments $N_G + 1$ to $N_S$ are calculated on CPU.

In the CPU calculation, the number of threads (~10) is usually smaller than the number of segments (>100). Each thread computes the likelihood of several segments, one by one, using Eq. (10). On the other hand, there are a large number of threads in a GPU (~1000), so it is more efficient to distribute photons rather than segments. In the GPU calculation, therefore, one block with $N_{GTh}$ threads calculates the likelihood of one segment. The photons are equally distributed to the threads, each of which performs matrix multiplication for assigned photons in Eq. (10) as

$$\mathbf{L}_j = \prod_{i=M \times j+2}^{M \times (j+1)+1} [\mathbf{F}(c_i) \exp(\mathbf{K}(t_i - t_{i-1}))]. \quad (28)$$

Here, $\mathbf{L}_j$ is the matrix in the likelihood computation of $M$ photons assigned to the $j$th GPU thread. Then, the likelihood value for a segment in Eq. (10) is calculated by multiplying the matrices in Eq. (28)

$$L = \mathbf{1}^T \prod_{j=0}^{N_{GTh}-1} \mathbf{L}_j \mathbf{F}(c_1) \mathbf{p}_{eq}. \quad (29)$$

After both CPU and GPU calculations are finished, the total log-likelihood is obtained by adding log-likelihood values of individual segments.

The parameter optimization was implemented in C/C++ using a multidimensional minimization function (gsl_multimin_fminimizer_nmsimplex2 of the GNU Scientific Library), which can be performed using a typical laboratory computer. For the parallelization, Windows multithreading API and CUDA 8.0 library were used and compiled by Visual Studio 2015 (Intel® Xeon® CPU E5-2620 v3, NVIDIA Quadro M2000).

**Correlation analysis of immobilization data**. A cross-correlation function of the data from the immobilization experiment was calculated as

$$C_{\alpha\beta}(\tau) = \frac{\overline{\langle N_\alpha(t + \tau) N_\beta(t) \rangle}}{\langle N_\alpha \rangle \langle N_\beta \rangle} - 1. \quad (30)$$

$N_\alpha(t)$ and $N_\beta(t)$ are the number of photons detected in $\alpha$ and $\beta$ channels in a bin at time $t$, $\langle \ldots \rangle$ is an average of a quantity in a given segment in a trajectory, and the upper bar indicates the average over segments. The correlation functions in Supplementary Fig. 6e were calculated for the segments longer than 10 ms.

**Reporting summary**. Further information on research design is available in the Nature Research Reporting Summary linked to this article.

## Data availability

All data supporting the main conclusion are included in the paper. Extra data are available from the corresponding author upon reasonable request. The source data underlying Figs. 4–6, and Supplementary Figs. 2b, c, 6e, 7–10, 11b–h are provided as a Source Data file. Source data are provided with this paper.

## Code availability

The maximum likelihood analysis software package, source code, and example data are available at https://github.com/hoisunglab/FRET_3colorCW. Source data are provided with this paper.

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

## Acknowledgements

We thank W.A. Eaton and A. Szabo for numerous helpful discussions and comments, A. Aniana for technical assistance, and J. Lloyd for mass spectrometry. This work was supported by the Intramural Research Program of the National Institute of Diabetes and Digestive and Kidney Diseases, NIH.

## Author contributions

J.Y., J-.Y.K., I.V.G, and H.S.C. designed the project and wrote the paper. J.Y. performed protein folding experiments and data analysis. J-.Y.K. expressed, purified, and labeled TAD and NCBD and performed binding experiments and data analysis. J.Y. implemented CPU–GPU parallelization. I.V.G. developed three-color FRET theory. J.M.L planned and prepared α₃D with three fluorophores.

## Competing interests

The authors declare no competing interests.
