## [Peer Review File · Nature Communications]

Reviewers' comments:

Reviewer #1 (Remarks to the Author):

The manuscript by Yoo et al. presents interesting technical advances in the acquisition and analysis of 3-color smFRET data, from a clean and elegant labeling process, optimized additives and excitation scheme, to a global maximum likelihood method accelerated by clever use of CPU-GPU processing. The authors illustrate their approach using examples from protein binding and folding kinetics down to the 1-millisecond timescale, and based on simulations even far into the sub-millisecond range. They demonstrate convincingly that their approach is superior to previous 3-color FRET implementations, both in terms of data quality and analysis. The results are consistent with previous work and at a very high technical level. Even though no new conceptual insights are presented, the combined technical advances will be of great value to practitioners of 3-color FRET and are likely to stimulate more activity in that field. The manuscript is written in a rather technical style, and given the methodological focus, this seems appropriate, although it might limit the potential target audience.

A point that might be confusing to some readers is the use of "ALEX" for pulsed interleaved excitation, which is more commonly abbreviated "PIE". Similarly, in the introduction, the authors imply that PIE has more commonly been used for 3-color FRET, but most published applications use cw excitation in combination with ALEX. It may be useful to clarify these points.

More information about the software or programming language used for the parallelization (and data analysis in general) would be interesting to have in the manuscript.

Reviewer #2 (Remarks to the Author):

In their manuscript titled "Fast three-color single-molecule FRET using statistical inference", Yoo et al. present an extension of the likelihood analysis developed by Gopich and Szabo for quantitative three-color FRET experiments on surface-immobilized molecules. The approach has the promise to reveal correlated conformational changes on the micro- to millisecond timescale. Notably, different to previous work published by the Chung lab¹, the use of continuous wave excitation alleviates some of the experimental difficulties, such as photobleaching and -blinking, by reducing the photophysical stress exerted on the fluorophores in pulsed experiments. To still acquire the required information to resolve all three FRET efficiencies (for which usually the FRET efficiency between the two acceptors has to be probed independently), the authors extend their analysis framework to utilize periods of the trajectory where only two of three fluorophores are active. Commendably, all analysis tools, including a fast implementation on GPUs using the CUDA framework, are made freely available with the publication. The framework is applied to two different experimental examples to study the folding/unfolding dynamics of α 3D and the binding dynamics of NCBD to the TAD domain. To this end, the authors also developed an approach to obtain specific labeling of all three fluorophores, an issue that has limited most prior protein studies utilizing 3cFRET.

The work is extensive, the method is well implemented, and the experiments are carried out correctly. However, I am missing the aspect that makes the use of three-color FRET necessary for the study of the experimental systems and justifies the increased complexity of the approach. It is thus questionable whether the paper would be appealing to the broader audience of the journal, and it might be better suited for a more specialized journal given the technical focus of the paper. For example, while the folding/unfolding dynamics of α 3D are quantified, no novel insights are obtained from having three labels on the protein. It would be more interesting to apply the method to a system that shows multi-step folding, where 3cFRET would enable to map the folding pathway in more detail, or to multi-domain proteins, where the sequential or simultaneous folding of the

individual domains could be revealed by 3cFRET. Likewise, the kinetic of NCBD binding to the TAD domain are quantified, but no statement is given on potential correlated conformational changes of TAD upon binding of NCBD, which would be the interesting aspect of the 3cFRET experiment.

The work is arguable the most complete framework for quantifying kinetics in three-color FRET to date and thus marks a significant step for the field. To appeal to the broader audience of the journal, however, I think it would be important that a model system is studied that highlights the unique advantages of three-color FRET to reveal correlated and coordinated motions. My detailed comments and questions are given below.

Major comments:

1. One potential problem that I see for the global treatment of two- and three-color segments is the assumption that the different segments the same principal FRET efficiencies are assumed to be observed for molecules with two or three active fluorophores. In other words, the analysis assumes that the same conformational states are visited irrespective of the presence or absence of either fluorophore. However, if we consider a molecule that assumes a vastly different conformation upon binding of an interaction partner (while potentially showing dynamics in the bound and unbound states), it would not be correct to use the two-color FRET efficiency obtained from the unbound state for the analysis of the three-color segments. In this way, the use of single-wavelength excitation sacrifices the aspect of the simultaneous measurement of all FRET efficiencies by global analysis of different segments in time, and correlated motions (one of the selling points of multicolor FRET studies) are potentially lost. I would be interested in the authors comments on this point.

2. On a similar note, the assignment of the FRET efficiencies of two-color segments to the states in the three-color segments should mainly rely on the kinetic signatures of the involved states. Could ambiguities potentially arise? How likely is it that the optimization terminates in a local minimum?

3. I understand that in the 3cCW experiments, it is not possible to plot the pairwise distributions of the different FRET efficiencies as they are not measured simultaneously. For the 3cALEX experiments, however, this should be possible. It would be interesting to investigate potential correlated motions in the studied systems and to visualize the different conformational states.

4. Previously, photon recoloring had been used to judge the fit quality of the obtained parameters, but similar analyses have not been performed here. How well do the extracted parameters explain the measured histograms?

5. The authors do not discuss the structural aspect of the extracted quantitative FRET efficiencies. How do the values compare to expected distances, e.g. from accessible volume calculations based on the structures shown in Fig. 2?

6. In their discussion of the deviations between 2c and 3c FRET efficiencies for the unfolded state of α 3D, the authors speculate that the differences may arise from the chain dynamics in the unfolded state. They use a Gaussian chain model to compute expected average FRET efficiencies, which however fails to account for the observed deviation. How do the inter-residue distances of the label positions compare to the parameters of the Gaussian chain (e.g. extracted from the 2cFRET efficiencies)? Given the failure to explain the deviation, and the fact that the deviation (in my judgement) is not that large, one could also think of moving the discussion into a supplementary note.

7. Are there heterogeneities between molecules? Do the extracted parameters vary if e.g. only a part of the dataset is analyzed?

8. In the study of the binding of NCBD to the TAD domain, how does the binding constant compare

to ensemble (reference) values? In addition to the change of the buffer ionic strength, it would have been interesting to perform a concentration series of NCBD as well. What was the concentration of NCBD in the given measurements?

9. Have controls on static systems been performed, e.g. on triple-labeled DNA or folded α 3D in native conditions? How would the authors distinguish a system showing fast conformational dynamics ($\sim\mu$ s) from a static system (both of which would result in quasi-constant FRET efficiency traces) using the likelihood formalism?

10. Have controls been performed assessing the effect of the surface attachment on the studied systems, e.g. by using solution-based measurements on the same setup?

11. Confidence intervals are specified for the rates in Fig. 4 (but not in Fig. 5) and Tables S1-2, but no uncertainties are given for the FRET efficiencies. As the authors used the curvature of the likelihood function to estimate the uncertainty, this information should also be available, and I suggest that it should be given for all parameters.

12. The approximate rates of acceptor dark-state formation could be measured by performing single-color experiments on molecules carrying only the acceptor fluorophores. Have such controls been performed? How do these rates compare to the rates obtained in the three-color analysis? Has the use of the more complex analysis model been justified using e.g. the Bayesian information criterion or a likelihood ratio test?

13. The kinetics could also be quantified by correlation analysis of the acquired single-molecule traces. Have the authors considered comparing the relaxation times obtained from FCS with the one obtained from the likelihood approach?

Minor comments:

14. The modality wherein the fluorophores are alternatingly excited by pulsed lasers is called "3cALEX" in the manuscript. The correct notation would (to the best of my knowledge) be 3cPIE 2 or alternatively ns-3cALEX3, as the acronym "ALEX" is generally used to describe alternating excitation schemes using cw laser (achieved e.g. through the use of AOTFs). I would advise to change the notation.

15. I tried to run the analysis of the test datasets provided in the GitHub repository but was unable to run the CUDA code on my PC using a RTX2080 GPU. I assume that this is an issue related to the need of re-compiling the MATLAB MEX files for the given architecture, but I had issue achieving this using the provided Visual Studio project. It would be beneficial if the authors could provide clear, step-by-step instructions for compiling the required binary files (aimed at users inexperienced with either VisualStudio or the CUDA framework).

16. A lot of space is designated to the CPU-GPU parallelization in Fig. 1e, but little description is given in the main text. Given the scope of the journal, I believe this part would be best shifted to the SI.

17. What is the time binning used for the FRET efficiency histograms shown in the main text?

18. The number of molecules should be specified in the main text figure captions. (Currently, this information is hidden in the SI.)

19. The authors use the term "acceptor fraction" to describe the signal fractions observed in the acceptor channels. This could potentially be confused with the fraction of molecules carrying an acceptor. I think these quantities would better be called "acceptor signal fractions".

20. Have the authors considered the use of a Bayesian approach to estimate the posterior

distribution of the fit parameters, e.g. using Metropolis-Hastings sampling?

1. Yoo, J., Louis, J. M., Gopich, I. V. & Chung, H. S. Three-Color Single-Molecule FRET and Fluorescence Lifetime Analysis of Fast Protein Folding. *J Phys Chem B* 122, 11702-11720 (2018).
2. Müller, B. K., Zaychikov, E., Bräuchle, C. & Lamb, D. C. Pulsed interleaved excitation. *Biophys J* 89, 3508-3522 (2005).
3. Kapanidis, A. N. et al. Alternating-Laser Excitation of Single Molecules. *Accounts of Chemical Research* 38, 523-533 (2005).

Reviewer #3 (Remarks to the Author):

In this manuscript, Yoo et al. develop a three-color FRET method using intense continuous-wave (CW) laser excitation instead of alternating laser excitation and achieve time resolution of protein folding and binding dynamics within milliseconds. They also develop analysis method using maximum likelihood of the donor and acceptor photons along the trajectories to resolve the kinetic parameters and fraction of states of the system. By parallelizing calculations of both CPU and GPU, they can largely accelerate data analysis. The manuscript is well written, especially considering the technical details required to derive this three-color FRET method. All code is provided in a public GitHub repository. The code is very well documented and commented, and the authors also provide a manual for installation and use of the programs. The repository is very well organized and should be sufficiently easy to follow for someone with coding experience. The organization of the code is professional quality.

However, there are few concerns on the validation and applicability of the method that I hope the author can address.

1. Although the authors compare the results from the new imaging and analysis method with the previous 2 color-FRET measurements and 3 color-ALEX measurements, I think it's necessary to perform the analysis on synthetic data with controlled variations, such as number of states, transition rates between biological states, photoblinking frequency, noisiness on the signal. With the synthetic data with known ground truth, it would give a better validation on the accuracy of the fitting.
2. The authors show two known systems, α 3D folding and p53-NCBD binding, which exist in only two biological states. Therefore, under such assumption, the authors build a 8-state model for the maximum likelihood analysis. However, for a system with multiple intermediate states, how accurate and how fast the analysis method can achieve. And for a system with unknown number of states, how to fit the number of states in the final parameter fitting?

Minor points:

1. Please provide the number of traces of each histogram.
2. The authors state that the maximization of the likelihood function takes "a long time" due to the large number of kinetic parameters that need to be optimized. This method requires access to many GPU and CPU cores, and presumably, the more GPU cores available the faster the derivation would occur. The authors should supply some estimate of the computational resources and time required for the analyses performed in this paper. Given that the analyses were performed for relatively simple systems (ones that had been previously characterized), one wonders if this method is feasible for any given lab. For example, how long does this maximization process take if a lab has access only to CPUs? Will computational resources be a limiting factor in this method?

Fig. S3 provides benchmark tests for single calculations and optimizations of the likelihood function, but does not state how many CPUs or GPUs were involved and how many total iterations are necessary.

3. In describing Figure 4a, the authors claim that the relaxation rates determined by the 3c-CW/maximum-likelihood method agree with previous measurements from 3c-ALEX experiments. However, the rates at GdmCl concentrations below 2.5 seem to disagree quite dramatically. There is some discrepancy between the figure and the description.

4. Fig. S1. Please provide the average photon counts and time before photobleaching in addition to the histograms for (a) and (b), respectively.

5. In Fig 3a. top figure, the authors claim the red arrow indicate the photobleaching of A2. However, there is another drop of A2 signal at the orange arrow which was not explained. Why is the red arrow position considered to be A2 photobleaching, rather than changing in FRET between A1 and A2?

6. The authors also point out during the α 3D denaturing, all three distances between fluorophores increase, however, this is not reflected by the two-color FRET in Fig.3c, which showed unchanged peak of DA1 and DA2 at different concentrations of GdmCl.

7. The author should quantify the ratio between bound state and unbound state at different concentrations of salt in Fig 5 as well.

8. Fig. 1d figure description states that "due to the binding to A2-labeled (middle) and A2-unlabeled (left) binding partners, three-color and DA1 part of the photon trajectories cannot be separated into different segments." This statement is a bit confusing, because in Fig 3d, binding event with labeled NCBD can be separated from the binding with unlabeled NCBD (the origin fragment).

9. In the 3c-CW, there are two ways to calculate E3c12 based on equation 7 and 8. Are the results the same using both ways?

Reviewer #1 (Remarks to the Author):

The manuscript by Yoo et al. presents interesting technical advances in the acquisition and analysis of 3-color smFRET data, from a clean and elegant labeling process, optimized additives and excitation scheme, to a global maximum likelihood method accelerated by clever use of CPU-GPU processing. The authors illustrate their approach using examples from protein binding and folding kinetics down to the 1-millisecond timescale, and based on simulations even far into the sub-millisecond range. They demonstrate convincingly that their approach is superior to previous 3-color FRET implementations, both in terms of data quality and analysis. The results are consistent with previous work and at a very high technical level. Even though no new conceptual insights are presented, the combined technical advances will be of great value to practitioners of 3-color FRET and are likely to stimulate more activity in that field. The manuscript is written in a rather technical style, and given the methodological focus, this seems appropriate, although it might limit the potential target audience.

→ We thank Reviewer #1 very much for considering our manuscript highly. We revised the manuscript according to reviewer's comments below.

1. A point that might be confusing to some readers is the use of "ALEX" for pulsed interleaved excitation, which is more commonly abbreviated "PIE". Similarly, in the introduction, the authors imply that PIE has more commonly been used for 3-color FRET, but most published applications use cw excitation in combination with ALEX. It may be useful to clarify these points.

→ Thank you for raising this issue. We agree that it may be confusing and revised the text in the introduction (page 3) as

"Typically three-color FRET is done with alternating laser excitation (ALEX)^{15,16} using CW lasers with intensity modulations or pulse-interleaved excitation (PIE)¹⁷ using pulsed lasers to determine all three FRET efficiencies (Fig. 1b)^{18,19}."

In addition to this clarification, since our previous work on α 3D folding was done using PIE, we replaced ALEX with PIE in the figure legends of Fig. 4 and related discussion.

2. More information about the software or programming language used for the parallelization (and data analysis in general) would be interesting to have in the manuscript.

→ More details and explanation on the software were included in the method section of the manuscript (page 32). Additional details of the software and analysis methods can be found in the manual uploaded manual with example data at https://github.com/hoisunglab/FRET_3colorCW.

"The parameter optimization was implemented in C/C++ using a multidimensional minimization function (gsl_multimin_fminimizer_nmsimplex2 of the GNU Scientific Library), which can be performed using a typical laboratory computer. For the parallelization, Windows multithreading API and CUDA 8.0 library were used and compiled by Visual Studio 2015 (Intel® Xeon® CPU E5-2620 v3, NVIDIA Quadro M2000)."

Reviewer #2 (Remarks to the Author):

In their manuscript titled “Fast three-color single-molecule FRET using statistical inference”, Yoo et al. present an extension of the likelihood analysis developed by Gopich and Szabo for quantitative three-color FRET experiments on surface-immobilized molecules. The approach has the promise to reveal correlated conformational changes on the micro- to millisecond timescale. Notably, different to previous work published by the Chung lab¹, the use of continuous wave excitation alleviates some of the experimental difficulties, such as photobleaching and -blinking, by reducing the photophysical stress exerted on the fluorophores in pulsed experiments. To still acquire the required information to resolve all three FRET efficiencies (for which usually the FRET efficiency between the two acceptors has to be probed independently), the authors extend their analysis framework to utilize periods of the trajectory where only two of three fluorophores are active. Commendably, all analysis tools, including a fast implementation on GPUs using the CUDA framework, are made freely available with the publication. The framework is applied to two different experimental examples to study the folding/unfolding dynamics of α 3D and the binding dynamics of NCBD to the TAD domain. To this end, the authors also developed an approach to obtain specific labeling of all three fluorophores, an issue that has limited most prior protein studies utilizing 3cFRET.

The work is extensive, the method is well implemented, and the experiments are carried out correctly. However, I am missing the aspect that makes the use of three-color FRET necessary for the study of the experimental systems and justifies the increased complexity of the approach. It is thus questionable whether the paper would be appealing to the broader audience of the journal, and it might be better suited for a more specialized journal given the technical focus of the paper. For example, while the folding/unfolding dynamics of α 3D are quantified, no novel insights are obtained from having three labels on the protein. It would be more interesting to apply the method to a system that shows multi-step folding, where 3cFRET would enable to map the folding pathway in more detail, or to multi-domain proteins, where the sequential or simultaneous folding of the individual domains could be revealed by 3cFRET. Likewise, the kinetic of NCBD binding to the TAD domain are quantified, but no statement is given on potential correlated conformational changes of TAD upon binding of NCBD, which would be the interesting aspect of the 3cFRET experiment.

The work is arguable the most complete framework for quantifying kinetics in three-color FRET to date and thus marks a significant step for the field. To appeal to the broader audience of the journal, however, I think it would be important that a model system is studied that highlights the unique advantages of three-color FRET to reveal correlated and coordinated motions. My detailed comments and questions are given below.

→ We are very grateful that Reviewer #2 considers that our development of the three-color method is a significant step for the field and that the experimental implementation is correct and well carried out. We agree that our applications still remain at the extraction of kinetic and FRET efficiency parameters for the simplest two-state systems as the first demonstration. However, we hope the reviewer appreciates that we have already put in tremendous effort to the development of the method itself including an enhanced computation strategy, which we believe will be a robust basis for the investigation of more complex systems in the future. In fact, we are searching for appropriate (complex folding) systems, to which we can apply our methods, but we anticipate it will take a fair amount of time to achieve the goals that the reviewer suggested. Instead, we added our analysis of simulated photon trajectories with a more

complex three-state model (one intermediate state between the bound and unbound states) also as response to the comments of Reviewer #3 (see Supplementary Fig. 11).

We also thank Reviewer #2 for careful reading and providing us with insightful comments. We revised the manuscript according to reviewer's comments and concerns as listed below. We hope that now our manuscript is suitable for publication in *Nature Communications*.

Major comments:

1. One potential problem that I see for the global treatment of two- and three-color segments is the assumption that the different segments the same principal FRET efficiencies are assumed to be observed for molecules with two or three active fluorophores. In other words, the analysis assumes that the same conformational states are visited irrespective of the presence or absence of either fluorophore. However, if we consider a molecule that assumes a vastly different conformation upon binding of an interaction partner (while potentially showing dynamics in the bound and unbound states), it would not be correct to use the two-color FRET efficiency obtained from the unbound state for the analysis of the three-color segments. In this way, the use of single-wavelength excitation sacrifices the aspect of the simultaneous measurement of all FRET efficiencies by global analysis of different segments in time, and correlated motions (one of the selling points of multicolor FRET studies) are potentially lost. I would be interested in the authors comments on this point.

→ In the analysis, we use the same kinetics model and kinetic model parameters (global fit) for three-color and two-color segments, but the FRET efficiency parameters are different for each state in two- and three-color segments. For example, in the two-state folding model, there are two acceptor fractions both for the folded and unfolded states in the three-color segments: (ϵ_{1F} , ϵ_{2F}) for the folded state and (ϵ_{1U} , ϵ_{2U}) for the unfolded state. For the two-color segments (e.g., donor and acceptor 1), ϵ_F^{DA1} and ϵ_U^{DA1} are the acceptor fractions (apparent FRET efficiencies in this case) for the folded and unfolded states, respectively. Then, two- and three-color parameters of the folded state, ϵ_{1F} , ϵ_{2F} , and ϵ_F^{DA1} are used to calculate the three FRET efficiencies (E_1 , E_2 , and E_{12}) of the folded states. The unfolded state parameters, ϵ_{1U} , ϵ_{2U} , and ϵ_U^{DA1} are used for the calculation of the three FRET efficiencies of the unfolded state. For the binding experiment, only bound state FRET efficiencies are obtained in this way because acceptor 2 is not present in the unbound state. To make it clear, the two-color FRET efficiencies obtained from the unbound state are not used for the analysis of the three-color part (i.e., bound state) as the reviewer pointed out. If one uses a more complex model, such as one intermediate state (I) between the bound and unbound states, the two-color and three-color acceptor fractions corresponding to state I are obtained and used for the calculation of the three FRET efficiencies as demonstrated in the new simulation (Supplementary Fig. 11). The method does not use parameters from vastly different conformations to calculate three FRET efficiencies, and therefore, it is possible to follow correlated motions by using an appropriate kinetics model.

2. On a similar note, the assignment of the FRET efficiencies of two-color segments to the states in the three-color segments should mainly rely on the kinetic signatures of the involved states. Could ambiguities potentially arise? How likely is it that the optimization terminates in a local minimum?

→ The reviewer is correct that the correlation between two- and three-color acceptor fractions rely on the kinetic signature. We don't think the optimization terminates in a local minimum for a two-state

system when the relative populations are clearly different, which is the case in most of our experimental conditions. However, when the relative population is close to 50:50 like the folding experiment of α_3D at 2.5 M GdmCl (F:U = 47:53), it may be possible that the folded and unfolded two-color parameters are correlated with the unfolded and folded three-color parameters, respectively, if we choose the initial parameters in that way (although there is no reason to do this). Even in this case, if necessary, we can utilize photobleaching of dyes to connect the same states with different dye combinations. For example, when acceptor 2 is photobleached in the three-color folded state, it will be immediately followed by the folded state with the donor and acceptor 1 in the trajectory. Therefore, it will be possible to avoid assigning the three-color folded state to the two-color unfolded state and vice versa. Actually, this is used in the analysis of the binding experiment because photobleaching of A2 has to be included in the kinetics model of the binding experiment.

3. I understand that in the 3cCW experiments, it is not possible to plot the pairwise distributions of the different FRET efficiencies as they are not measured simultaneously. For the 3cALEX experiments, however, this should be possible. It would be interesting to investigate potential correlated motions in the studied systems and to visualize the different conformational states.

→ The reviewer is correct that the correlated motions cannot be directly visualized in the 3cCW scheme because three-color and two-color segments cannot be collected simultaneously for the same molecules. This is possible in 3cALEX (PIE) experiments, but only when the dynamics are slower than the time resolution (bin time). For changes on a fast time scale (comparable to or faster than the bin time) in our systems, this visualization is not possible as demonstrated in our previous 3c-PIE study of α_3D folding (Yoo *et al.* J Phys Chem B 122, 11702-11720 (2018)). In this case, maximum likelihood photon trajectory analysis should be employed with a kinetic model anyway. Direct visualization of the correlated dynamics should be possible for other slow folding/binding systems in the near future.

4. Previously, photon recoloring had been used to judge the fit quality of the obtained parameters, but similar analyses have not been performed here. How well do the extracted parameters explain the measured histograms?

→ We added the comparison of the recolored histograms with the experimental histograms in Supplementary Fig. 6. The results show excellent agreement between the recolored and experimental histograms except for two-color DA2 histogram of the binding experiment. This discrepancy may result from the very low peak height of the bound state. This is discussed in the text and figure legend.

Main text (page 11),

“The histograms constructed from recolored photon trajectories using maximum likelihood parameters (8-state model)²² are very similar to the experimental histograms (Supplementary Fig. 6), confirming the good accuracy of the extracted parameters.”

Main text (page 14),

“Similar to α_3D experiment, the histograms constructed from the recolored photon trajectories using the extracted parameters (8-state model) agree well with the experimental histograms (Supplementary Fig. 6).”

Supplementary Fig. 6,

“The histograms constructed from the recolored photon trajectories using the extracted parameters (8-state model) agree well with the experimental histograms, except overestimation of the bound state peak of the DA2 histograms in (d). This discrepancy may result from the inaccuracy of the determination of the slow blinking rates of A2 (k'_{b2} and k'_{d2} , see Supplementary Table 2) and overall very low fraction of the bound state peak.”

5. The authors do not discuss the structural aspect of the extracted quantitative FRET efficiencies. How do the values compare to expected distances, e.g. from accessible volume calculations based on the structures shown in Fig. 2?

→ The TAD/NCBD structure shown in Fig. 2 is one of the NMR structures and several residues at both N- and C-termini of TAD are flexible. Therefore, it would not be easy to calculate the average distance and FRET efficiency directly from the structure. In addition, the chemical structure of acceptor 2, CF680R is unknown. Therefore, the only distance that can be estimated from the structure is between the donor (Alexa 488) at the N-terminus and A1 (Alexa 594) attached to residue 33 of α_3D . We tried accessible volume calculation developed by Seidel and coworkers (Nat. Methods 9, 1218-1225 (2012)) for residue 1 and 33 of α_3D (PDB id: 2A3D). This is an approximation because the donor was attached to the cysteine residue appended to the N-terminus of the protein. The FRET averaged distance is 4.23 nm, which corresponds to the FRET efficiency of 0.81 ($R_0 = 5.4$ nm). This value agrees very well with our measurement, 0.81 – 0.85 at three different GdmCl concentrations (Supplementary Table 1). We added this discussion on page 12,

“Compared to E_2^{3c} , E_1^{3c} values are stable. The folded state FRET efficiency E_{1F} from all three measurements agree very well (Fig. 4c). These values are also close to the FRET efficiency of 0.81 calculated from the distance distribution of D and A1 attached to residue 1 and 33 obtained from the accessible volume calculation²⁸ using the protein structure (PDB id: 2A3D, Fig. 2).”

6. In their discussion of the deviations between 2c and 3c FRET efficiencies for the unfolded state of α_3D , the authors speculate that the differences may arise from the chain dynamics in the unfolded state. They use a Gaussian chain model to compute expected average FRET efficiencies, which however fails to account for the observed deviation. How do the inter-residue distances of the label positions compare to the parameters of the Gaussian chain (e.g. extracted from the 2cFRET efficiencies)? Given the failure to explain the deviation, and the fact that the deviation (in my judgement) is not that large, one could also think of moving the discussion into a supplementary note.

→ The inter-residue distance between the labeled positions is unknown for the unfolded state (no structural information available). Therefore, there is nothing to compare with the distance distribution (and its mean value) obtained from the Gaussian chain model. The discussion is relatively short (part of one paragraph), so we hope to keep this in the main text unless the reviewer considers it is necessary to move this to Supplementary Note.

7. Are there heterogeneities between molecules? Do the extracted parameters vary if e.g. only a part of the dataset is analyzed?

→ From the previous work with a well-characterized two-state system, protein G with very slow kinetics, for which individual states can be assigned without ambiguity, we are aware that the FRET efficiency distribution is wider than the width by shot noise (Chung et al., PNAS v106, 11837 (2009)). Therefore, we expect similar heterogeneity in the current systems as well, but by analyzing data from ~100 molecules, parameters would converge to their average values. We tested this by dividing the data of the binding experiment into two sets (all three NaCl concentrations) and re-analyzing, which results in similar parameters in general. The fitting results are compared in Supplementary Table 3.

8. In the study of the binding of NCBD to the TAD domain, how does the binding constant compare to ensemble (reference) values? In addition to the change of the buffer ionic strength, it would have been interesting to perform a concentration series of NCBD as well. What was the concentration of NCBD in the given measurements?

→ The concentrations of NCBD at different NaCl concentrations are listed in Supplementary Table 2. At each NaCl concentration, it is expected that the apparent binding rate increases linearly with the NCBD concentration. The bulk dissociation constant of the complex has been measured using ITC by Jemth and coworkers (Biochemistry v54, 4741 (2015)). At the ionic strength of 44 and 74 mM, the dissociation constants were 3 and 13 μM , which agree very well with our previous two-color measurement ($K_d = 1.6 - 11 \mu\text{M}$) at similar ionic strength of 38 – 98 mM (Nat Commun, v9, 4707 (2018)). We added this discussion on page 14 as

“... the dissociation constant K_d from our single-molecule measurements are comparable with those from ensemble measurements (isothermal calorimetry) by Jemth and coworkers²⁹. At the ionic strength of 44 and 74 mM, the dissociation constants were 3 and 13 μM , which agree very well with our previous two-color measurement ($K_d = 1.6 - 11 \mu\text{M}$) at similar ionic strength of 38 – 98 mM¹³.”

As for the three-color result, as mentioned in the text (page 14), the bound complex is destabilized by the second acceptor A2 on NCBD at low salt concentrations (ionic strength of 8 – 38 mM) in the three-color experiment, but the difference is not so large compared to the values from two-color experiments. The calculated dissociation constants are now plotted in Fig. 5b and listed in Supplementary Table 2.

9. Have controls on static systems been performed, e.g. on triple-labeled DNA or folded $\alpha_3\text{D}$ in native conditions? How would the authors distinguish a system showing fast conformational dynamics ($\sim\mu\text{s}$) from a static system (both of which would result in quasi-constant FRET efficiency traces) using the likelihood formalism?

→ The dynamics probed in this manuscript is ~ 1 ms and therefore, the signature of two-state dynamics is observed in the binned histograms with a bin time of 1 ms (Fig. 3b and 3e). There are clear shifts of the peaks for both ε_1 and ε_2 for $\alpha_3\text{D}$ upon denaturation (Fig. 3b) and the bound state peaks at $\varepsilon_1 \sim 0.1$ and 0.6 for TAD/NCBD binding clearly disappear by increasing NaCl concentration. We have not performed an experiment for $\alpha_3\text{D}$ at the native condition, but from this trend, it is easy to predict that the

histograms will be similar to those at 2 M GdmCl with a single narrow peak in all histograms without tails towards the unfolded state.

If the dynamics is much faster than the bin time, the histogram would show a single peak in all cases as seen in our previous folding study of villin headpiece (Chung *et al.* Chem Phys v422, 229 (2013)). The relaxation rate was $30 - 50 \text{ ms}^{-1}$. In this case, the dynamics can also be detected by cross-correlation analysis of the donor and acceptor intensity and the amplitude and relaxation time can be compared with the parameters from the maximum likelihood analysis as described in the two-color work. If the narrow distribution results from a single state, the maximum likelihood method will detect only acceptor blinking as shown in our previous study of amyloid- β (Meng *et al.*, Biophysical Journal, vol 114, 870 (2018)) (in this case, proteins are in the completely unfolded state). The correlation analysis would be more complex in three-color experiments especially for the binding experiment, but can be done for folding experiment (see comment #13).

10. *Have controls been performed assessing the effect of the surface attachment on the studied systems, e.g. by using solution-based measurements on the same setup?*

→ We collected solution-based data (free-diffusion experiment) for both systems. For $\alpha_3\text{D}$, both one-dimensional (for two-color trajectories) and two-dimensional (for three-color trajectories) histograms are similar to those of the immobilization experiments, assuring the surface immobilization effect is small. This was also tested in our previous experiment using two-color FRET (Chung *et al.*, JPCA, v115, 3642 (2011)). The kinetic parameters agree well with those from the three-color experiment in the current work. For binding of TAD and NCBD, on the other hand, gamma factor is quite low for A2 ($\gamma_2 = 0.49$), which results in the suppression of bursts from the bound state by lowering photon counts below the photon threshold (50 photons per 2 ms bin). We observed the bound state peak disappears in the histogram (Supplementary Fig. 4d and e). Therefore, the direct comparison would not be possible for the binding experiment. Nevertheless, the similar dissociation constants obtained from the bulk measurement and from the current work (comment #8) suggest the surface effect on the dynamics is small. We added discussion on pages 8 and 11 and Supplementary Fig. 4.

“We also performed free-diffusion experiments, in which molecules are not immobilized, but freely diffuse and emit a burst of fluorescence photons when they pass through the laser focus. The acceptor fraction histograms are shown in Supplementary Fig. 4 for the comparison with the results from the immobilization experiment. The one- and two-dimensional acceptor fraction histograms from the free-diffusion experiments are similar to those from the immobilization experiments (Supplementary Fig. 4), which indicates the surface immobilization effect on the protein dynamics is small as also verified in the previous two-color FRET study²⁵.”

“The two-dimensional acceptor fraction histograms from the free-diffusion and immobilization experiments are compared in Supplementary Fig. 4. Compared to $\alpha_3\text{D}$ folding experiment, γ -factor of A2 ($\gamma_2 = 0.49$) is much lower, and the number of photons detected in a bin (2 ms) is less than the threshold (50 photons) for most bursts from the bound state. Therefore, the bound state peak is not observed in the histograms and a direct comparison between the free-diffusion and immobilization experiments is not possible. However, the dissociation constant measured in this work is similar to that from an ensemble binding experiment, which indicates the surface immobilization effect is very small (see below for more discussion).”

11. *Confidence intervals are specified for the rates in Fig. 4 (but not in Fig. 5) and Tables S1-2, but no uncertainties are given for the FRET efficiencies. As the authors used the curvature of the likelihood function to estimate the uncertainty, this information should also be available, and I suggest that it should be given for all parameters.*

→ The confidence intervals for the acceptor fractions can be directly obtained from the curvature of the likelihood function. However, the FRET efficiencies are obtained after complex conversion process and it is not straightforward to calculate the errors. We estimated errors by simple error propagation by assuming that the errors of each variables are independent. This would overestimate the errors of the FRET efficiencies, but these errors would not be so large compared to the real errors because the errors of the acceptor fractions are very small. We added the error bars for the rate constants in Fig. 5a and listed the errors for all the parameters in Supplementary Table 1 and 2. However, we did not plot error bars of the FRET efficiencies in the figures because they are very small. During the inspection of the errors, we found that the fitting was terminated (it didn't reach the maximum of iteration, though) before finding the likelihood maximum for the binding experiment. Therefore, we replaced the data in Fig. 5 and Supplementary Table 2. All parameters are very similar except for the blinking rate of acceptor 1. In addition, we performed new simulation with the correct parameters (Fig. 6b).

12. *The approximate rates of acceptor dark-state formation could be measured by performing single-color experiments on molecules carrying only the acceptor fluorophores. Have such controls been performed? How do these rates compare to the rates obtained in the three-color analysis? Has the use of the more complex analysis model been justified using e.g. the Bayesian information criterion or a likelihood ratio test?*

→ First, we currently don't have a laser that directly excite acceptor 2 (680 nm), and therefore cannot measure A2 blinking by direct excitation. To compare blinking of A1 from the three-color experiment and from A1-only trajectories by direct A1 excitation, we analyzed the data collected previously with 3cPIE (Yoo et al. J Phys Chem B 122, 11702 (2018)). The blinking parameters of A1-only trajectories were obtained by the maximum likelihood method using the photon count rate assuming Poissonian photon statistics (Gopich and Szabo, JPCB 113, 10965 (2009)). The rate from the dark to the bright state of A1 (k_{b1}) are comparable: 5.97 (2 M), 4.71 (2.25 M), and 5.35 ms^{-1} (2.5 M) from the three-color measurement and 2.45 (2 M), 3.32 (2.25 M), and 3.64 ms^{-1} (2.5 M) from A1-only trajectory analysis. However, the bright state populations (p_{b1}) from the A1-only trajectory analysis are smaller than those from the three-color measurement: 0.99 (2 M), 0.99 (2.25 M), and 0.99 (2.5 M) from the three-color measurement and 0.93 (2 M), 0.94 (2.25 M), and 0.95 (2.5 M) from A1-only trajectory analysis. It is not clear what causes this discrepancy. It is possible that the photon count statistics is not Poissonian and we sometimes observe fluctuations and drift in the count rate in a trajectory. In addition, the presence of additional fluorophores in FRET experiments can change the dye property. Nonetheless, Bayesian information criterion (BIC) is much smaller for the model with acceptor blinking, which justifies the use of models including acceptor blinking (Supplementary Table 1 and 2).

13. *The kinetics could also be quantified by correlation analysis of the acquired single-molecule traces. Have the authors considered comparing the relaxation times obtained from FCS with the one obtained from the likelihood approach?*

→ For three-color trajectories, it is not straightforward how to calculate the correlation function because there are three channels. For α_3 D folding, the correlation functions were calculated for the D and A1 channels for two-color DA1 trajectories and for A1 and A2 channels and D and A2 channels for three-color trajectories (see Supplementary Fig. 6e). Overall, the correlation amplitude is small due to the small difference between the acceptor fractions of the folded and unfolded states. Therefore, we fit only the data at 2.25 M GdmCl, where the amplitude is the largest due to the comparable folded and unfolded state population. The relaxation rates of the three correlation functions are $\sim 1.5 \text{ ms}^{-1}$, which is in a reasonable agreement with the value obtained from the maximum likelihood method (~ 1.4 -fold difference). As for the binding data, however, calculating correlation function is more complex. Since TAD can bind to both A2-labeled and A2-unlabeled (or inactive) NCBD, which results in completely different patterns of photon count rate changes. For example, A1 count rate will decrease when TAD binds to A2-labeled NCBD, but will increase when it binds to unlabeled NCBD. In addition, photobleaching of A1 will also affect the correlation analysis. Therefore, correlation analysis would not provide any quantitative comparison with the likelihood analysis.

We added a section in Methods that describes the cross-correlation analysis and Supplementary Fig. 6e.

Minor comments:

14. *The modality wherein the fluorophores are alternatingly excited by pulsed lasers is called “3cALEX” in the manuscript. The correct notation would (to the best of my knowledge) be 3cPIE 2 or alternatively ns-3cALEX3, as the acronym “ALEX” is generally used to describe alternating excitation schemes using cw laser (achieved e.g. through the use of AOTFs). I would advise to change the notation.*

→ Thank you for pointing this. We revised the text in the introduction (page 3) as

“Typically three-color FRET is done with alternating laser excitation (ALEX)^{15,16} using CW lasers with intensity modulations or pulse-interleaved excitation (PIE)¹⁷ using pulsed lasers to determine all three FRET efficiencies (Fig. 1b)^{18,19}.”

In addition to this clarification, since our previous work on α_3 D folding was done using PIE, we replaced ALEX with PIE in the figure legends of Fig. 4 and related discussion.

15. *I tried to run the analysis of the test datasets provided in the GitHub repository but was unable to run the CUDA code on my PC using a RTX2080 GPU. I assume that this is an issue related to the need of re-compiling the MATLAB MEX files for the given architecture, but I had issue achieving this using the provided Visual Studio project. It would be beneficial if the authors could provide clear, step-by-step instructions for compiling the required binary files (aimed at users inexperienced with either VisualStudio or the CUDA framework).*

→ We are very grateful that the reviewer actually spent time to install and test the analysis code. We added a step-by-step instruction for downloading and installing necessary software (section 5.2.2 in the revised software manual).

16. *A lot of space is designated to the CPU-GPU parallelization in Fig. 1e, but little description is given in the main text. Given the scope of the journal, I believe this part would be best shifted to the SI.*

→ Fig. 1e is moved to Supplementary Fig. 2a.

17. *What is the time binning used for the FRET efficiency histograms shown in the main text?*

→ The bin time is 1 ms and we state this at the beginning of Fig. 3 legend.

18. *The number of molecules should be specified in the main text figure captions. (Currently, this information is hidden in the SI.)*

→ The number of analyzed molecules is now included in the figure captions of the main text (Fig. 3).

19. *The authors use the term “acceptor fraction” to describe the signal fractions observed in the acceptor channels. This could potentially be confused with the fraction of molecules carrying an acceptor. I think these quantities would better be called “acceptor signal fractions”.*

→ A more accurate term would be “fraction of acceptor photon count rate”. Therefore, we used this term as much as possible along with “acceptor fraction” in the manuscript. In addition, since we have already used this in the previous manuscript, we would like to keep the same terms for consistency. We use labeling efficiency for the fraction of molecules carrying acceptor 2, which we believe is not confusing. We also put the definition more clearly at its first appearance in the last paragraph of the introduction (page 5).

“Importantly, we found that the FRET efficiency conversion from the extracted fractions of acceptor photon count rates (acceptor fraction, ε) can be inaccurate depending on the specific set of values of a system.”

The acceptor fractions are also defined and described in the section “Fractions of acceptor photon count rate and FRET efficiency” in Results (pages 6 and 7).

20. *Have the authors considered the use of a Bayesian approach to estimate the posterior distribution of the fit parameters, e.g. using Metropolis-Hastings sampling?*

→ Given the large number of parameters (14 and 16 parameters), it would take too long time to cover the entire dimension of parameter space. Therefore, we prefer not to perform this calculation.

Reviewer #3 (Remarks to the Author):

In this manuscript, Yoo et al. develop a three-color FRET method using intense continuous-wave (CW) laser excitation instead of alternating laser excitation and achieve time resolution of protein folding and binding dynamics within milliseconds. They also develop analysis method using maximum likelihood of the donor and acceptor photons along the trajectories to resolve the kinetic parameters and fraction of states of the system. By parallelizing calculations of both CPU and GPU, they can largely accelerate data analysis. The manuscript is well written, especially considering the technical details required to derive this three-color FRET method. All code is provided in a public GitHub repository. The code is

very well documented and commented, and the authors also provide a manual for installation and use of the programs. The repository is very well organized and should be sufficiently easy to follow for someone with coding experience. The organization of the code is professional quality.

However, there are few concerns on the validation and applicability of the method that I hope the author can address.

→ We are very grateful that the reviewer thinks highly of our development of the method and appreciates our effort to make the analysis code easily accessible to the community. We revised the manuscript according to reviewer's comments and concerns below and hope that this is suitable for publication in *Nature Communications*.

1. Although the authors compare the results from the new imaging and analysis method with the previous 2 color-FRET measurements and 3 color-ALEX measurements, I think it's necessary to perform the analysis on synthetic data with controlled variations, such as number of states, transition rates between biological states, photoblinking frequency, noisiness on the signal. With the synthetic data with known ground truth, it would give a better validation on the accuracy of the fitting.

→ We actually analyzed synthetic data for a two-state system with a wide range of relaxation rates and different populations both for folding and binding experiments to explore how fast dynamics can be probed by this method. The results are displayed in Fig. 6 and Supplementary Fig. 10. The simulation results show that the accuracy of population is excellent and the relaxation rate can be extracted within 5 – 10 % error even when the rate is close to the photon count rate. Acceptor fractions are now shown in Supplementary Fig. 10, which are very accurate in general. Nevertheless, it is an important point to demonstrate the performance of the method for a system with more than two protein conformational states. Therefore, we simulated photon trajectory data with a three-state model for binding (bound-intermediate-unbound) and the result is summarized in Supplementary Fig. 11. However, there are so many parameters that can be varied and it would be too time-consuming to analyze photon trajectories to explore the effect of all variables (we simulated 5 photon trajectories for each parameter set). Therefore, we fixed the intermediate population (10%) and varied only relaxation rates and bound state population as in the two-state model simulation. We believe more complete and realistic simulation can be performed when a true multi-state system is investigated.

2. The authors show two known systems, α 3D folding and p53-NCBD binding, which exist in only two biological states. Therefore, under such assumption, the authors build a 8-state model for the maximum likelihood analysis. However, for a system with multiple intermediate states, how accurate and how fast the analysis method can achieve. And for a system with unknown number of states, how to fit the number of states in the final parameter fitting?

→ As mentioned above, we simulated the data with a three-state model. Then, we analyzed the data using two-state and three-state models to explore the difference. The fitting errors of the parameters and deviations from the input simulation values are shown in Supplementary Fig. 11. There are 22 parameters for three-state model and 16 parameters for the two-state model and it takes about 5.5 times longer to find three-state parameters than two-state parameters. The number of states is usually determined by comparing Bayesian information criterion (BIC) or similar criteria. As an example, we compared BIC of the two-state and three-state model analysis in Supplementary Fig. 11h. BIC is clearly smaller for the three-state model compared to the two-state model. However, when the number of states

are unknown for a real experimental system, one should be cautious to use statistical criteria such as BIC to determine the number of states. In real experimental data, there are many factors that can cause the system look more complex than expected such as heterogeneity in the immobilization microenvironment and fluorescence signal from impurity, which can be accounted for better by using more molecular states in a model. Therefore, it would be prudent to use any prior knowledge from other experimental methods rather than blindly trust statistical criteria from the analysis.

Minor points:

1. *Please provide the number of traces of each histogram.*

→ We added this information in the caption of Fig. 3 and Supplementary Table 3.

2. *The authors state that the maximization of the likelihood function takes “a long time” due to the large number of kinetic parameters that need to be optimized. This method requires access to many GPU and CPU cores, and presumably, the more GPU cores available the faster the derivation would occur. The authors should supply some estimate of the computational resources and time required for the analyses performed in this paper. Given that the analyses were performed for relatively simple systems (ones that had been previously characterized), one wonders if this method is feasible for any given lab. For example, how long does this maximization process take if a lab has access only to CPUs? Will computational resources be a limiting factor in this method? Fig. S3 provides benchmark tests for single calculations and optimizations of the likelihood function, but does not state how many CPUs or GPUs were involved and how many total iterations are necessary.*

→ The calculation was performed with a single CPU and two GPU that are used in typical desktop computers. Intel® Xeon® CPU E5-2620 v3 processor has 6 cores (12 logical cores) and NVIDIA Quadro M2000 GPU has 768 cores, which are not high-end processors at all. Since the additional acceleration by GPU is about 25% (Supplementary Fig 2), the optimization using only CPU will still work very well. The number of iterations for the data in Supplementary Fig. 2c is ~ 1,000 for folding. The number of iterations is larger (~ 5 times) for binding data analysis probably because of the more complex kinetic model and slower convergence of the parameters due to inclusion of the rare photobleaching events of acceptor 2 in the model. We added a statement in the text that the analysis can be performed using a typical laboratory computer (page 32).

“The parameter optimization was implemented in C/C++ using a multidimensional minimization function (gsl_multimin_fminimizer_nmsimplex2 of the GNU Scientific Library), which can be performed using a typical laboratory computer. For the parallelization, Windows multithreading API and CUDA 8.0 library were used and compiled by Visual Studio 2015 (Intel® Xeon® CPU E5-2620 v3, NVIDIA Quadro M2000).”

3. *In describing Figure 4a, the authors claim that the relaxation rates determined by the 3c-CW/maximum-likelihood method agree with previous measurements from 3c-ALEX experiments. However, the rates at GdmCl concentrations below 2.5 seem to disagree quite dramatically. There is some discrepancy between the figure and the description.*

→ The relaxation rates at 2 and 2.25 M GdmCl are $1.24 (\pm 0.06)$ and $0.91 (\pm 0.03)$ ms^{-1} in the previous 3c-ALEX (PIE) experiment and $1.16 (\pm 0.07)$ and $1.05 (\pm 0.03)$ ms^{-1} in the current work. We think this is a very good agreement.

4. *Fig. S1. Please provide the average photon counts and time before photobleaching in addition to the histograms for (a) and (b), respectively.*

→ The mean values of the distributions are included in the caption of Supplementary Fig. 1.

5. *In Fig 3a. top figure, the authors claim the red arrow indicate the photobleaching of A2. However, there is another drop of A2 signal at the orange arrow which was not explained. Why is the red arrow position considered to be A2 photobleaching, rather than changing in FRET between A1 and A2?*

→ This is an excellent point. Thank you for your careful reading. The signal in A2 channel between red (A2 bleach) and orange (A1 bleach) arrows are due to the leak of A1 photons into A2 channel. The long tail of A1 fluorescence spectrum penetrates the emission filter cutoff for A2. Therefore, the decrease of A2 signal at the position where the orange arrow indicates is due to A1 photobleaching. This point is now explained in the caption of Fig. 3a.

“In the upper trajectory, A2 fluorescence after A2 photobleaching (red arrow) that is maintained before A1 photobleaching (orange arrow) results from the leak of A1 photons into A2 channel.”

In addition, we put a red arrow for A1 photobleaching in the lower trajectory of Fig. 3a by mistake. If this is A2 photobleaching, there should be a large increase of A1 signal (folded state). Therefore, this should be an orange arrow as we corrected.

6. *The authors also point out during the α 3D denaturing, all three distances between fluorophores increase, however, this is not reflected by the two-color FRET in Fig.3c, which showed unchanged peak of DA1 and DA2 at different concentrations of GdmCl.*

→ There are very small decreases in both DA1 and DA2 histograms (Fig. 3c) with increasing GdmCl concentrations. The change is very small mainly due to the small difference (~ 0.1) of two-color acceptor fractions between the folded and unfolded states. Please see $\epsilon_{F,U}^{\text{DA1}}$ and $\epsilon_{F,U}^{\text{DA2}}$ values in Supplementary Table 1. We revised the text to include this explanation on page 8,

“The shifts of the histograms of two-color trajectories (DA1 and DA2, Fig. 3c) are much smaller primarily due to the small difference of acceptor fractions ϵ^{DA1} and ϵ^{DA2} between the folded and unfolded states. (see accurate determinations of parameters using the maximum likelihood method in the following sections).”

7. *The author should quantify the ratio between bound state and unbound state at different concentrations of salt in Fig 5 as well.*

→ The ratio of the bound and unbound states does not result in a meaningful quantity because this depends on the concentration of the binding partner NCBD. Instead, the ratio of the unbound to bound

populations multiplied by NCBD concentration is the dissociation constant K_d . We included this in Fig. 5b and Supplementary Table 2.

8. *Fig. 1d figure description states that “due to the binding to A2-labeled (middle) and A2-unlabeled (left) binding partners, three-color and DA1 part of the photon trajectories cannot be separated into different segments.” This statement is a bit confusing, because in Fig 3d, binding event with labeled NCBD can be separated from the binding with unlabeled NCBD (the origin fragment).*

→ The reviewer is correct. In Fig. 3d, one can separate three-color and DA1 part of the photon trajectory only when the molecule is in the bound state (in the case of slow binding). However, the unbound state is always two-color, three-color and DA1 part of the trajectories must be analyzed together for the kinetics analysis. To avoid confusion, we revised the text in the caption of Fig. 1d as

“The three-color and DA1 parts of the photon trajectories cannot be separately analyzed as in the case of folding in (c) because the unbound state is always two-color (DA1), whereas the bound state is either three-color or two-color (DA1) due to the binding to either A2-labeled (middle) or A2-unlabeled (left) binding partner.”

9. *In the 3c-CW, there are two ways to calculate E3c12 based on equation 7 and 8. Are the results the same using both ways?*

→ They are similar as shown in Fig. 5e and Supplementary Table 2.

REVIEWERS' COMMENTS:

Reviewer #2 (Remarks to the Author):

All my comments and suggestions have been addressed with remarkable details and efforts. I have no further remarks other than congratulate the authors on this milestone for three-color FRET. I am looking forward to the applications of this new tool to more complex systems (and potentially my own research).

Regarding the author's comment to point 6 concerning the discussion of the Gaussian chain model (lines 353-372), I have no objections to keep this part in the main text.

Reviewer #3 (Remarks to the Author):

In the revised manuscript, the reviewers have addressed all my previous comments. One additional suggestion: it is worth adding the response to my major point #2 in terms of model selection for unknown system to the discussion.

Reviewer #2 (Remarks to the Author):

All my comments and suggestions have been addressed with remarkable details and efforts. I have no further remarks other than congratulate the authors on this milestone for three-color FRET. I am looking forward to the applications of this new tool to more complex systems (and potentially my own research).

Regarding the author's comment to point 6 concerning the discussion of the Gaussian chain model (lines 353-372), I have no objections to keep this part in the main text.

→ **Thank you very much!**

Reviewer #3 (Remarks to the Author):

In the revised manuscript, the reviewers have addressed all my previous comments. One additional suggestion: it is worth adding the response to my major point #2 in terms of model selection for unknown system to the discussion.

→ **We added the following paragraph in "Detection of intermediate state" section (page 10).**

“When the number of states is unknown, it is usually determined by comparing statistical criteria such as Bayesian information criterion (BIC). In Supplementary Fig. 11h, BIC is clearly smaller for the three-state model compared to the two-state model. However, one should be cautious to use statistical criteria such as BIC to determine the number of states. In real experimental data, there are various factors that can cause the system look more complex (i.e., more states) such as heterogeneity in the immobilization microenvironment and fluorescence signal from impurity. Therefore, it would be prudent to use any prior knowledge from other experimental methods rather than blindly trust statistical criteria from the analysis.”